

# Calibrating Climate Models Using Inverse Methods: Case studies with HadAM3, HadAM3P and HadCM3

Simon F. B. Tett[1], Kuniko Yamazaki[1,2], Michael J. Mineter[1], Coralia Cartis[3], and Nathan Eizenberg[3,4]

[1]School of Geosciences, University of Edinburgh, Crew Building, Alexander Crum Brown Road, The King's Buildings, Edinburgh EH9 3FF. UK
[2]Met Office, Fitzroy Road, Exeter,Devon, EX1 3PB. UK
[3]Mathematical Institute, University of Oxford, Andrew Wiles Building, Radcliffe Observatory Quarter, Woodstock Road, Oxford, OX2 6GG. UK
[4]Bureau of Meteorology, GPO Box 1289, Melbourne, VIC 3001, Australia

*Correspondence to:* Simon Tett (simon.tett@ed.ac.uk)

**Abstract.** Optimisation methods were successfully used to calibrate parameters in an atmospheric component of a climate model using two variants of the Gauss-Newton line-search algorithm. 1) A standard Gauss-Newton algorithm in which, in each iteration, all parameters were perturbed. 2) A randomized block-coordinate variant in which, in each iteration, a random sub-set of parameters was perturbed. The cost function to be minimized used multiple large-scale observations and was constrained to produce net radiative fluxes close to those observed. These algorithms were used to calibrate the HadAM3 (3rd Hadley Centre Atmospheric Model) model at N48 resolution and the HadAM3P model at N96 resolution.

For the HadAM3 model, cases with seven and fourteen parameters were tried. All ten 7-parameter cases using HadAM3 converged to cost function values similar to that of the standard configuration. For the 14-parameter cases several failed to converge, with the random variant in which 6 parameters were perturbed being most successful. Multiple sets of parameter values were found that produced multiple models very similar to the standard configuration. HadAM3 cases that converged were coupled to an ocean model and ran for 20 years starting from a pre-industrial HadCM3 (3rd Hadley Centre Coupled model) state resulting in several models whose global-average temperatures were consistent with pre-industrial estimates. For the 7-parameter cases the Gauss-Newton algorithm converged in about 70 evaluations. For the 14-parameter algorithm with 6 parameters being randomly perturbed about 80 evaluations were needed for convergence. However, when 8 parameters were randomly perturbed algorithm performance was poor. Our results suggest the computational cost for the Gauss-Newton algorithm scales between $P$ and $P^2$ where $P$ is the number of parameters being calibrated.

For the HadAM3P model three algorithms were tested. Algorithms in which seven parameters were perturbed and three out of seven parameters randomly perturbed produced final configurations comparable to the standard hand tuned configuration. An algorithm in which six out of thirteen parameters were randomly perturbed failed to converge.

These results suggest that automatic parameter calibration using atmospheric models is feasible and that the resulting coupled models are stable. Thus, automatic calibration could replace human driven trial and error. However, convergence and costs are, likely, sensitive to details of the algorithm.





# 1  Introduction

Weather and Climate models need to parametrise unresolved processes (Edwards, 2011) and representation of these processes often contain parameters which have a broad range of plausible values (Murphy et al., 2004; Stainforth et al., 2005). Tuning or calibration of climate models by finding parameter combinations or introducing new processes is rarely well documented, often

time consuming, and the metrics used opaque (Mauritsen et al., 2012; Hourdin et al., 2013) with the main approach being trial and error. Consequently, expensive person time is needed to calibrate or tune climate models. Methods that could automatically calibrate model parameters would allow easier development of parametrisations, allow objective discussion of the observed targets and allow more rapid development of climate models. Such an approach would also facilitate uncertainty analysis and would improve understanding the contribution of parametrisation compared to resolved dynamics in model properties including

model error.

Tett et al. (2013b) (T13 from here on) outlined an approach to model parameters calibration by considering it as an inverse optimisation problem for which the aim is to find the parameter values which produce an atmospheric model with smallest error relative to a predetermined set of weighted observations. T13 focused on only two observations: global mean outgoing longwave and reflected shortwave radiation, and modified only four parameters. They were able to calibrate the model pa-

rameters to several different observational targets. In this paper we further develop the approach taken by T13 to increase the number of observations and parameters used. We then couple some of the resulting atmospheric models to an ocean model to test if the resulting coupled model is stable.

Various approaches have been taken to optimising model parameter values. Golaz et al. (2013) hand tuned the GFDL CM3 model to radiation balance by adjusting several parameters in the cloud scheme parameters in the cloud scheme finding a

significant impact on aerosol forcing but not on greenhouse gas forcing or on "Cess" climate sensitivity (Cess et al., 1990). They found very large differences during the 20th century due to the perturbed impact of aerosols. Bellprat et al. (2012, 2015) generated a model emulator for three climate variables from a regional model. From this emulator by latin-hypercube sampling they found the parameter combinations that minimised error. Their earlier work focused on 5 parameters while their recent paper used 8 parameters and considered a North American and European regions. They found the calibrated model improved

the simulation of summers in both regions. Williamson et al. (2013) use a combination of emulation and ruling out implausible observations to construct models. They used four observational constraints: global average surface air temperature (SAT), northern hemisphere meridional temperature gradient, seasonal cycle of temperature in the Northern Hemisphere and global average precipitation. They found that SAT was the most important constraint. Latter they included the strength of the Antarctic Circumpolar Current (ACC) in their analysis and found parameter combinations where the model had a good simulation of

both the ACC and SAT (Williamson et al., 2014).

Irvine et al. (2013) generated 200 variants of HadCM3 using a Latin hypercube experimental design and splitting each parameter range into 200 bins. They then ran the resulting coupled models and found that about 10% were acceptable. Tomassini et al. (2015), using a low resolution version of the MPI-ESM model, perturbed 8 parameters randomly across their plausible range and generated coupled models with a broad range of global average temperatures. They then examined the different feed-





backs and mechanisms for those feedbacks in their model finding that four convective parameters related to convective mixing had strong impacts on both the mean tropical circulation and on climate sensitivity. Such brute force approaches become extremely expensive as the dimensionality of the problem increases though the use of emulators may help.

Attempts have been made using data assimilation techniques to calibrate parameters. Such systems simultaneously estimate the atmospheric state and the parameter values. Schirber et al. (2013) reported on a study in which they used that approach but found no improvement in the model climatology. Ruiz and Pulido (2015) used a similar algorithm and found an improvement in medium-range forecast skill but did not report on the impact on model climatology.

Another approach is to use forecast error. Ollinaho et al. (2012) updated four parameter values and their covariances iteratively using a set of three day forecasts of ECHAM5 and found a modest reduction in forecast error. When they ran the model with observed sea surface temperature and sea-ice they found a reduction in top of atmosphere flux errors. They followed up this study with one in which they minimised forecast errors in the total energy (Ollinaho et al., 2014). They also applied the technique to the ECMWF forecasting system and found a modest change in parameter values and an increase in forecast skill in the tropics (Ollinaho et al., 2013).

The approach we consider is optimisation via direct evaluation of the model something attempted by Jones et al. (2005) for a low resolution version of HadCM3. Yang et al. (2013), building on Jackson et al. (2004), applied the SSAA algorithm to tune parameters in CAM5 to improve the simulation of the partititioning between convective and large-scale precipitation. Zou et al. (2014) applied a similar approach to an East Asian regional model modifying seven parameters and optimising only mean precipitation. They found a significant improvement in both the rainfall pattern and daily rainfall distribution.

Here we update T13 to include a larger number of observations and parameters. As before we continue to use a Gauss-Newton algorithm but include a randomized block-coordinate variant where, on each iteration, a random sub-set of the parameters are perturbed.

Our objectives are:

1. Test how well a Gauss-Newton algorithm does in minimising error in the HadAM3 N48 model (Pope et al., 2000) with 7 and 14 parameters, and multiple observations.

2. Test for eqifinality in which models with different parameter values have similar observed values (Beven and Freer, 2001).

3. See how coupled model variants of HadCM3 (Gordon et al., 2000) with the parameters taken from the optimisation behave.

4. Test these algorithms with the N96 HadAM3P model (Massey et al., 2015).

The reminder of this paper first describes the models we use, the optimisation method and observational metrics used. We next describe results of optimisation, the properties of the atmospheric models and how the coupled models behave. We discuss our results before concluding.





## 2 Methods

In this section we outline our methods. We first describe two related atmospheric models we use. Next we outline the Gauss-Newton algorithm, and a randomized block-coordinate variant of it, deal with the need to regularize matrices, and how the algorithm terminates. We then describe the choices we made in parameter selection and parameter perturbation, the observa-
tions and covariance matrices we used. Finally we describe how we evaluate the optimised configurations.

### 2.1 Models

We use the N48 ($3.75° \times 2.5°$) resolution configuration of HadAM3, which uses a 360 day calendar, driven with the same package of forcings used by T13. Simulations were ran from 1st Dec 1998 to 1st March 2005 (6.25 years) and the period 1st March 2000 to 30th February 2005 compared with observations. In addition we use the N96 ($1.875° \times 1.25°$) configuration
of HadAM3P (Massey et al., 2015) with a similar package of forcings as used in the N48 configuration. This model was ran from 1st Dec 1999 to 1st March 2005 (5.25 years) We use the standard land-surface dataset rather the time-varying dataset used in the N48 case, include both the direct and indirect effect of $SO_2$ aerosols on clouds (Jones et al., 2001) and used, after interpolation, the same ozone dataset us we used in the N48 case. Some results from the default configuration are described in Tett et al. (2013a).

### 15 2.2 Gauss-Newton and line-search

We build on the approach used by T13 which minimized an objective function which was the root mean square of the global average outgoing longwave radiation and reflected shortwave radiation. We extend this to a larger number of observations taking account of both observational error and simulated internal variability. As we focus on large scale, multi-annual averages we assume that both terms can be represented by multi-variate Gaussian distributions characterised by covariance matrices $C_O$
(observational error) and $C_i$ (internal variability) respectively. If the model was perfect we would expect $(S - O) \sim N(0, C)$ where $C = C_O + 2C_i$. Therefore, the cost-function ($F(p)$) depending on parameters ($p$) we minimise is:

$$F^2(p) = \frac{(S - O)^T C^{-1} (S - O)}{N} \tag{1}$$

where $N$ is the number of observations. This requires that $C$ is invertible and, if necessary, we regularize it (see below).

This way of defining $F(p)$ allows for covariance between observations to be taken account of. For example internal vari-
ability might generate large correlation between total outgoing radiation and precipitation and so not weighting them would give greater weight to configurations with small error in outgoing radiation and precipitation than is justified. We also want to reduce the importance of observations with high uncertainty and, conversely, increase the weight of observations with small uncertainty. We follow Sexton and Murphy (2011) and generally use a crude estimate of observational error based on the difference between two different observational datasets. Our aim in this paper is the application of inverse methods to pa-





rameter calibration not on producing good estimates of observational error. That is a matter for the groups that produce the observational datasets and so beyond the scope of the work reported on here.

We estimate $C_i$ from 100 simulations of the standard N48 HadAM3 model configuration. Estimating observational error is more difficult. For the radiation observations we use the fractional error estimates from Loeb et al. (2009) and apply them to

each regional value. For other datasets we define them as the difference between the default values and the equivalent from another observational dataset. We explored applying a covariance structure to the observational error but found this did not work well (see discussion) nor was there very strong objective justification for any covariance structure.

The Gauss-Newton algorithm is an iterative two-step algorithm. The first step is to compute the Jacobian $\mathcal{J}$

$$\mathcal{J}_{ij} = \frac{\partial S_i(\boldsymbol{p})}{\partial p_j} \tag{2}$$

where $S_i(\boldsymbol{p})$ is the $i$th simulated observation when the model is run with the vector of parameters $\boldsymbol{p}$ and $p_j$ is the $j$th parameter. We approximate this using finite differences (Nocedal and Wright, 2006):

$$\boldsymbol{J_{ij}} = \frac{S_i(\boldsymbol{p}+\Delta p_i \boldsymbol{e}^i) - S_i(\boldsymbol{p})}{\Delta p_i} \tag{3}$$

with $\Delta p_i$ being a suitably small perturbation to the $i$th parameter and $\boldsymbol{e}^i$ is the $i$th coordinate vector $(0,\dots,0,1,0\dots,0)$. In order to avoid using parameters values outside the expert range we chose, at each iteration, the sign of $\Delta p_i$ so as to perturb

towards the middle of the allowed range. Note that $\Delta p_i$ is ideally chosen such that the Jacobian is above internal variability and not, as is common, to machine precision. The choice of $\Delta p_i$ follows our noise estimates and use ideas from implicit filtering techniques for derivative free optimisation (Nocedal and Wright, 2006, Chapter 9) Having computed the Jacobian, the algorithm proceeds by computing the line-search vector ($\boldsymbol{s}$) to proceed along to minimize the cost function, $F$, through solving the linear problem:

$$\boldsymbol{Hs} = \boldsymbol{J}^T \boldsymbol{C}^{-1} (\boldsymbol{O} - S(\boldsymbol{p})) \tag{4}$$

where $\boldsymbol{H} = \boldsymbol{J}^T \boldsymbol{C}^{-1} \boldsymbol{J}$ is the finite difference approximation to the Hessian matrix ($\mathcal{H} = \mathcal{J}^T \boldsymbol{C}^{-1} \mathcal{J}$).

Having computed the line-search vector, $\boldsymbol{s}$, we then evaluate $F^2(\boldsymbol{p})$ at several steps along it ("Line-search"). If any of the chosen "line-search" parameter values are outside the expert defined plausible range we project these to the appropriate boundary. The minimum value of $F^2$ from the line search is used as the starting point for the next iteration.

The Gauss-Newton algorithm can be modified to include an additional constraint by modifying the cost function to:

$$F^2(\boldsymbol{p}) = \frac{\left((\boldsymbol{S}-\boldsymbol{O})^T \boldsymbol{C}^{-1} (\boldsymbol{S}-\boldsymbol{O})\right) + \frac{1}{2\mu}(O_c - S_c)^2}{N+1} \tag{5}$$

where $O_c$ and $S_c$ are the values we wish to constrain, and $\mu$ is an user choice to be decided on after experimentation This can be rewritten in the same form as Eqn. 1 with $\bar{\boldsymbol{O}} = (\boldsymbol{O}, O_c)$ , $\bar{\boldsymbol{S}} = (\boldsymbol{S}, S_c)$ and $\bar{\boldsymbol{C}} = \begin{pmatrix} \boldsymbol{C} & 0 \\ 0 & 2\mu \end{pmatrix}$



Building on ideas of Nesterov (2012) and Kim and Lee (2008) we also tried a randomized block-coordinate version of Gauss-Newton in which, each iteration, $P_{\mathrm{rand}}$ different parameters were chosen at random and used in both the Gauss-Newton and Line-search steps. Non perturbed parameters used the values from the previous iteration.

### 2.3 Scaling and Regularisation

Our algorithm could suffer from using ill-conditioned matrices in two places.

First, if the Hessian matrix is singular or ill-conditioned, defined as having a condition number greater than $10^{10}$, we use a Tihkonov regularisation (Nocedal and Wright, 2006) in which we add a small multiple of an identity matrix to the Hessian matrix. We iteratively increase the identity matrix scaling by a factor of ten starting with $10^{-7}$ until the regularized Hessian is no longer ill-conditioned or the scaling is $10^{-2}$. In the latter case our algorithm terminates with an error. This regularisation in-

troduces a scale dependence into the algorithm. Each time we compute the Jacobian we scale all parameters whose magnitudes are less than 1 so they have magnitude 1 and invert this scaling when computing the line-search direction.

Secondly, we also regularize $\boldsymbol{C}$. Rather than adding the identity matrix we scale the diagonal of the covariance matrix by increasing factors of two until the condition number of the entire matrix is less than five times the condition number of the diagonal matrix. We apply this regularisation after scaling all values and before computing the Jacobian. For the bulk of our

work $\boldsymbol{C}$ is well conditioned so this regularization is not applied.

### 2.4 Algorithm Termination

We need criteria to terminate the algorithm. Classical Gauss-Newton terminates when sufficiently close to the stationary point of the cost-function ($F(\boldsymbol{p})$) and so $F$ stops reducing (Nocedal and Wright, 2006). However, the climate is a chaotic system which introduces noise into the model evaluations. Therefore, the algorithm may continue to iterate even when it is not making

any significant progress or terminate because of not improving due to this noise.

The algorithm terminates on iteration $k$ when:

1. $F(\boldsymbol{p_k}) - F(\boldsymbol{p_{k-1}}) < c$ where $\boldsymbol{p_k}$ are the parameter values at iteration $k$. That is $F(\boldsymbol{p})$ has not reduced by a critical amount $c$.

2. $2(\boldsymbol{S_k} - \boldsymbol{S_{k-1}})\boldsymbol{C_i}^{-1}(\boldsymbol{S_k} - \boldsymbol{S_{k-1}})^T/N \leq c_i$ where $\boldsymbol{S_k}$ is the simulated observations at iteration $k$ and $c_i$ is a critical value

from a $\chi^2$ distribution with $N$ degrees of freedom. This checks that the new and previous simulated observations ($\boldsymbol{S}$) are statistically similar.

3. $(\boldsymbol{S_k} - \boldsymbol{O})\boldsymbol{C}^{-1}(\boldsymbol{S_k} - \boldsymbol{O})^T/N \leq c_o$ where $c_o$ is a critical value from a $\chi^2$ distribution with $N$ degrees of freedom. This checks that the current simulated observations are in statistical agreement with the target observations.

In our implementation $c$, $c_i$ and $c_o$ are all choices to be made in the algorithm.

For the random variant of the algorithm if the cost function did not reduce by $c$ then algorithm was restarted from the previous best parameter set by rerunning that case and another set of random perturbations. If the error then failed to reduce





by $c$ the algorithm would terminate. This means that the random variant will require at least two iterations before it terminates. This approach results in some duplicate simulations though, because of model chaos, the simulated observations differ. Some inefficiency results from this which could be reduced by keeping track of all cases that have run and not rerunning those cases. For ease of implementation we did not do this. Future work could implement such an optimisation.

## 2.5 Parameter selection and step size

We used up to fourteen parameters from the analysis of Yamazaki et al. (2013) but restricted to parameters that varied continuously. Some of those parameters are "meta-parameters" in that changes in one affected other parameters. We used the same algorithms as Williamson et al. (2013) did to modify parameters from the meta-parameters. Ranges of allowed parameter values were taken from Murphy et al. (2004).

We carried out three cases:

1. We adjusted seven parameters using HadAM3. Step sizes for the Jacobian calculation were taken from T13 for ENTCO-EFF, VF1, CT and RHCRIT. For the remaining three parameters we used 10% of their range.

2. We adjusted fourteen parameters, again, using HadAM3. To compute the step size for the additional parameters we set the value to the upper or lower range value that was most different from the standard value. Then for all 14 parameters, computed $d_i = (\boldsymbol{S}(\Delta p_i) - \boldsymbol{S_o})\boldsymbol{C_i}^{-1}(\boldsymbol{S}(\Delta p_i) - \boldsymbol{S_o})^T$. Where $d_i$ was greater than 100 we reduced $\Delta p_i$ by approximately $\sqrt{d_i/100}$. And where $d_i$ was less than 100 we increased $\Delta p_i$, limiting the increase to 50% of the allowed range, so that $d_i$ would, assuming linearity, be greater than 100.

3. We adjusted seven and thirteen parameters using HadAM3P using the same step sizes as in the fourteen parameter HadAM3 cases.

Parameters, ranges, default values and step sizes for the Jacobian computations are shown in Table 1.

## 2.6 Observations, Covariance matrices and optimisation choices

Here we describe the choices we made in our optimisation study,

We focus on large scale properties of the climate system and so consider the Northern Hemispheric extra-tropical ($\theta > 30°$N), tropical ($30°$S $\geq \theta \leq 30°$N) and the Southern Hemispheric extra-tropical ($\theta < 30°$S) means. We do this for seven variables:

**Land Air Temperature (LAT)** Land temperature has an impact on simulated biology, evaporation, snow and other important parts of the Earth System with changes in it being a significant impact from climate change. We use the observed CRU TS Vn 3.21 dataset (Harris et al., 2014), the HadAM3 N48 land/sea mask to determine land, and restrict to data north of $60°$S. For a second estimate of LAT we use ERA-Interim data (Dee et al., 2011).

**Land Precipitation (LP)** This is a key measure of the hydrological cycle. We also use the CRU TS Vn 3.21 dataset, the HadAM3 N48 land/sea mask, and restrict to data north of $60°$S. For a second estimate we use the vn6 GPCC dataset (Schneider et al., 2011). All simulated and observed values were converted to mm/day.



**Mean Sea Level Presure (SLP)** We use this as a measure of the planetary scale circulation. To correct for model mass loss we used sea-level pressure differences between the global-average value and the extra-tropical Northern hemisphere and tropics. We did not include the southern extra-tropics as that provided no new information and consequently made the covariance matrix uninvertable. We used values from ERA-Interim as observations and, for a second estimate used, the NCEP reanalysis (Kalnay et al., 1996). All observations and simulations were converted to hPa.

**Reflected Shortwave Radiation (RSR)** This measures the reflectivity of the Earth and is driven by clouds, snow, sea-ice and other surface properties. We compute values, and uncertainties, from the vn2.8 EBAF dataset (updated from Loeb et al. (2009)).

**Outgoing Longwave Radiation (OLR)** This is a measure of the outgoing thermal radiation from the Earth and is driven by atmospheric temperatures and clouds. We also use the vn2.8 EBAF dataset.

**Temperature at 500 hPa (T500)** This gives an estimate of the temperature lapse rate. We use ERA-Interim data as observations and for a second estimate use the NCEP reanalysis.

**Relative Humidity at 500 hPa (q500)** This provides a measure of mid-troposphere water vapour which is an important greenhouse gas. We also estimate values from ERA-Interim and use the NCEP reanlaysis as a second estimate.

See table 2 for target values used in all our studies.

We need to estimate a total covariance matrix ($C$) and a covariance matrix for internal variability ($C_i$). We estimated observational uncertainty for each regional OLR and RSR from the fractional uncertainties in Loeb et al. (2009). For other datasets we estimated the standard deviation as the difference between two different datasets. We assumed no correlation in observational error so $C_o$ is diagonal. The diagonal values of $C_o$ were significantly larger than the equivalent values of $C_i$ (internal variability) so observational error is the dominant term in the total error-covariance matrix ($C$).

We also applied a constraint (see sub-section 2.6) in order to generate atmospheric models that had a net radiative flux close to the observed value. After some experimentation we settled on a value of $\mu$ of 0.01 corresponding to an observational error of 0.015 Wm$^{-1}$ close to the observational error of about 0.2 Wm$^{-1}$ that Tett et al. (2013c) estimated from the difference of observational datasets.

When producing the datasets for the 7 parameter cases we made two errors in the computation of $C$. First we computed it as $C_i + C_O$ and secondly we miss-specified the three precipitation components. Given the focus of our work was on optimisation rather then the exact definition of the cost-function we don't believe these errors are very significant.

## 2.7 Evaluation

We evaluate the inverse approach in several different ways. For the algorithm we consider the expected number of iterations, evaluations and final error following the approach of T13 of using a strategy of repeatedly running the Gauss-Newton algorithm after it failed until convergence. This gives the expected number of model evaluations ($E$):

$$E = E_c + E_f \frac{f}{1 - f} \tag{6}$$





where $E_c$, $E_f$ and $f$ are the mean number of evaluations (or simulations) for studies that were comparable to, or better than, the standard configurations, the mean number of evaluations for studies that failed and the fraction that failed respectively. The expected number of iterations is computed similarly except iterations rather than simulations is used.

The line-search component of the algorithm has a selection effect as it takes the parameter combination that produced the smallest cost-function. Due to chaos in the model which leads to pseudo-random noise this will lead to a selection effect as the smallest cost-function values may have arisen by chance. To avoid this effect and to examine the properties of the resulting models we take the successful parameter sets and for each one run an ensemble of two simulations from December 1998 to April 2010. We compare results of these independent simulations for 2000-2005 with the standard configuration and each other and look for evidence of equifinality (Beven and Freer, 2001) in which different parameter combinations produce models that appear similar. . For greater out-of-sample comparison we compare differences between the 2005-2010 and 2000-2005 periods from the observations we use, the standard and independent optimised simulations.

For the HadAM3 cases we also carry out 20-year simulations of HadCM3 (Gordon et al., 2000) using the converged parameter sets. We compare results from the last 10 years of the 20-year simulation with the standard Control simulation of HadCM3 all started from the same initial state corresponding to about 5000 years of spinup.

# 3 Results

In this section we present our results. We tried several different algorithms using the HadAM3 and HadAM3P atmospheric models. We first present numerical results on the convergence behaviour of those algorithms, then compare some aspects of the climatologies of the modified models with the standard mode. Finally, we report on results of variants of the coupled atmosphere-ocean HadCM3 model that uses the optimal parameter sets from the HadAM3 test cases.

## 3.1 Atmospheric Model Convergence

We carried out several case studies. The first one in which we perturbed seven parameters using the Gauss-Newton algorithm. Using 14 parameters we tested the Gauss-Newton algorithm and two random parameter variants. Finally we tested three algorithms using the HadAM3P model configuration. In no case did the algorithm terminate because the cost-function was small. Given the crudeness of the observational covariance used in our cost-function we don't draw any inference from this. That would require a much better estimate of observational error than we made. Instead we take the pragmatic view that a perturbed model is comparable to (substantially better than) the standard configuration, in the simulation of the observations we used, if the cost function is less than 120 (80) % of standard models' cost function. We stress that this is a subjective choice that we made.

### 3.1.1 HadAM3 7-Parameter case

For the 7-parameter (HadAM3-7) trials we generated twelve random initial parameter choices by selecting values from their extreme limits (Table 3). For this algorithm we tried out five line-search evaluations at scalings of 1.0, 0.7 ,0.3, 0.1 and 0.01 of





the search vector and required $F(\boldsymbol{p})$ to reduce to keep iterating (i.e $c = 0$). Two cases failed in the first iteration with a model error with the remaining ten cases terminated when they failed to make progress. All those ten had cost values similar to the default model's value of 5.0 (Fig. 1(a)). These cases took between 3 and 12 iterations to terminate. As in our earlier study (T13) the cost-function reduces rapidly over the first one to two iterations with slow reduction after that (Fig. 1(a)).

5  We carried out five line-searches partially to test if any of the scalings on the search-vector were preferred. We found no strongly preferred scaling value (Table 4). In the rest of the paper we use scalings of 1, 0.7 and 0.3 on the search vector.

### 3.1.2 HadAM3 14-Parameter cases

We trialled three related algorithms to perturb 14 parameters. The algorithms we tested were the standard Gauss-Newton algorithm (HadAM3-14) and two variants with random perturbations. In one we perturbed 6 random parameters (HadAM3-10 14r6) and the other 8 (HadAM3-14r8). For each algorithm we did five studies with each one being started from the same random extreme parameter choices (Table 3). As described above we corrected the error in the computation of $C$ and adjusted the parameter perturbations (Table 1). For the random variants we required that the cost function reduce by 0.2 to continue iterating. Many of the simulations failed due to being marginally unstable. In which case we perturbed parameters by about one part in 1000 and reran that case. An operational system would restart the model with a small perturbation to a previous 15 state and run past the failure point.

  Unlike the HadAM3-7 cases the HadAM3-14 cases did not all produce cost function values comparable to the default model (Fig. 1(b)) with three cases failing and two succeeding. The successful cases took between four (74) and six (108) iterations(evaluations) with the unsuccessful cases taking one to four iterations. Neither of the successful cases are obviously better than the standard configuration.

20  Next turning to the HadAM3-14r6 cases. This algorithm performed well with four out the five cases succeeding taking between 6 (60) and 9 (87) iterations (evaluations). Three of the cases had cost functions less than the standard configuration but not substantially so (Fig. 1(b)). In contrast the HadAM3-14r8 algorithm performs poorly with only one case having a cost-function comparable with the standard configuration. This case took 5 (61) iterations (evaluations) to terminate. The unsuccessful cases took four iterations to terminate.

25 ### 3.1.3 HadAM3P cases

The HadAM3P cases differ from the standard configuration not only in increased resolution but in the addition of a cloud anvil parametrisation and the indirect effects of aerosols on cloud optical properties (Massey et al., 2015). One approach to model development would be to take the parameters from the previous model version and then re-calibrate the parameters using inverse methods with the new model. We tested three algorithms with all cases starting from the default HadAM3 parameters. 30 Our comparison case is the default HadAM3P configuration.

  Unless stated otherwise all studies used the same choices of covariance matrices, observations, parameter perturbations and other choices as the HadAM3 14 parameter studies (Table 1). So, for example, at each iteration the cost-function would need to reduce by 0.2 for the algorithm to continue. The three algorithms were:





**HadAM3P-13r6** The diffusion parameter was kept at its default HadAM3P values but all remaining 13 parameters were changed with six being chosen, at random, in each iteration.

**HadAM3P-7** Here the same parameters as used in the HadAM3 7-parameter cases were perturbed and termination occurred immediately the cost-function did not decrease by 0.2

**HadAM3P-7r3** As HadAM3P13r6 but with, at each iteration, 3 parameters, of the seven used in the 7 parameter HadAM3 case, perturbed at random.

The standard configuration of HadAM3P (Fig. 1(c)) is substantially worse, using our metric, than the standard HadAM3 configuration (Fig. 1(b)). Starting from the standard parameters the cost-function reduces less than for the HadAM3 cases which all started from extreme parameter choices. The HadAM3P-7 and HadAM3P-7r3 cases produced configurations comparable

with the standard HadAM3P model. The HadAM3P-7r3 study took 5 iterations with 31 evaluations. The HadAM3-7 case took 3 iterations also needing 31 evaluations of the model. The HadAM3P-13r6 case failed to converge and needed 3 iterations to fail.

### 3.1.4   Algorithm Performance

For each algorithm we tested using HadAM3 we characterise its performance using Eqn. 6. For each of the three HadAM3P

algorithms we only carried out one case so algorithm performance is evaluated from that single case.

As discussed earlier there is a potential selection effect in that from the line-search evaluations we chose the one case with minimum error. To examine the effect of this we compared the average cost from the optimised cases with the independent runs and with the cost values for the standard cases. Note that the independent and optimised cases have identical parameter sets but the 14- and 7-parameter algorithms use slightly different cost functions. The mean cost from the independent simulations

is, except for the HadAM3P-7r3 algorithm, larger than the mean cost for the optimised simulations (Table 5). The mean difference between the independent and best optimisation depends on the algorithm but ranges from 0.2 to 0.6 (5 to 15%) of the cost function for the standard configurations.

The expected number of iterations increases from the HadAM3-7 to HadAM3-14 algorithms but does not double. Our earlier work (T13) found that the median number of iterations for optimisation using two observations and four parameters required

between three and five iterations. This suggests that the cost of increasing the number of parameters is not excessive with the iteration count increasing less than $P$ (the number of parameters). As each iteration needs $P$ model evaluations then the total number of iterations likely increases between $P$ and $P^2$.

The six random parameter (HadAM3-14r6) algorithm worked well with an average cost function slightly better than the standard configuration (Table 5). Though requiring 60% more iterations than the 7-parameter case it only has an additional

20 % more expected evaluations for twice as many parameters. Random selection of 6 parameters has many less expected evaluations than perturbing all 14 parameters on each iteration. However, perturbing 8 parameters at random performs very much worse than perturbing 6 at random or all 14 parameters. We will explore possible reasons for this later. For the HadAM3P cases the HadAM3P-13r6 algorithm failed which while both the HadAM3P-7r3 and HadAM3P-7 algorithms succeeded.





To summarize this subsection we find that a relatively simple Gauss-Newton algorithm works well to automatically cali-brate parameters in an atmospheric model. The algorithm did not reduce the error to zero and so terminated when it stopped improving. We found that the expected number of iterations increases, though less than linearly, as we increased the number of parameters. Random selection of 6 out of 14 parameters worked well though random selection of 8 from 14 worked poorly.

We were also able to reduce the cost function of the HadAM3P model relative to the standard configuration of that model.

## 3.2 Atmospheric Model Evaluation

We now investigate the behaviour of the optimised HadAM3 and HadAM3P models by first focusing on the optimal parameters, then examining the simulation of the target observations in the independent simulations before comparing the model fields of key variables with observations. We aim to test for equifinality (Beven and Freer, 2001) where different parameter sets can lead

to very similar outputs. This could arise from multiple minima or a single broad flat minima.

We normalise the parameter values by their expert-based plausible ranges with 0 being the minimum and 1 the maximum. We find for both the 7- and 14-HadAM3 case studies that many of the parameters have a broad range of optimal values (Fig. 2). For each parameter we test if the distribution of optimal values is significantly different from a 0-1 uniform distribution using a Kolmogorov-Smirnov test. For three parameters (RHCRIT, ENTCOEFF and CW_LAND), in the 7-parameter case, we can reject

this null hypothesis. For the 14 parameter cases we can reject the null hypothesis of a uniform distribution for the same three parameters and, in addition, an additional five parameters have distributions inconsistent with a uniform distribution. These results suggest that minimizing the cost function does provide a weak constraint on some individual parameters.

We now consider how the independent simulations behave for the successfully optimised HadAM3-7- & -14, and HadAM3P parameter sets. These, to remind the reader, are two simulations ran with the same parameter set as the successful optimised

case. All model observation differences are normalised by the diagonal elements of the covariance matrix which is dominated by our crude estimate of observational error.

For the HadAM3-7 & -14 cases the optimised simulations are, for many target observations, similar to the standard config-uration (Fig. 3) with little scatter across the best cases. The 14-parameter cases have larger scatter than the 7-parameter cases suggesting the additional parameters lead to more ways to produce an optimised model. The medians are generally, though

not always, a small improvement (closer to zero) than the standard cases. However, for the optimised and standard parameter sets several simulated observations are outside the $\pm 2\sigma$ uncertainty range suggesting that further model improvement would need better representation of processes either through new parametrisations or higher resolution. Reflected shortwave radiation biases show greatest variation across the optimised cases with Northern Hemisphere extra-tropical land air temperature and topical RH at 500 hPa also showing large variation across the optimised cases.

Turning to the two optimised HadAM3P cases. These configurations have, like the standard HadAM3P, smaller biases in land air temperature across the three large regions we consider. This is particularly so in the NH extra tropics suggesting that enhanced resolution improves this particular observation. However, this model has a much worse simulation of precipitation in the tropics, even with tuning, than does the HadAM3 case. Optimising the parameters does reduce biases in the HadAM3P model but not enough to support the claim that is better than its lower resolution and computational cheaper HadAM3 cousin.





Comparison of the optimised cases with the initial random parameter choices gives a sense of how important variation in the parameters is for those observational biases. One thing that stands out is that large scale biases in the tropics (Fig. 4) are sensitive to parameter values. In contrast biases in extra-tropical relative humidity at 500 hPa are insensitive to changes in parameter values suggesting this is driven by the large scale resolved dynamics rather than parameterisation. In the extra-

tropics biases in RSR and OLR are the most sensitive to parameter variation with temperature at 500 hPa, MSLP and NH precipitation being least sensitive. That would suggest that the behaviour of these latter variables are mainly driven by the large scale resolved dynamics rather than the parametrisations.

We now examine how the bias changes when we consider a period outside the period we used to calibrate the model. Here we compare changes in bias between March 2005 – February 2010 and March 2000 – February 2005. We normalise by the

expected internal variability. For most observations and optimised configurations the bias does not significantly change between the two periods (Fig. 5) with the standard configurations and optimised cases behaving similarly. However, the extra-tropical relative humidity show significant changes in bias between the two periods with all simulations showing a significant increase in bias. As all models behave similarly this suggests either a lack of homogenisation in the ERA-Interim reanalysis or some systematic bias in all models.

So far we have focused on large scale biases. We use Taylor diagrams (Taylor, 2001) to examine how fields from the independent simulations compare with the observations. We focus on the same fields and observational datasets as used to compute the biases described above. Taylor diagrams summarize field similarity by computing field correlations and centred field standard deviations. We use the normalised variant where the centred field standard deviations are scaled by the equivalent values from the observed field we use. This allows us to compare fields with different units.

We find that for land air temperature, 500 hPa temperature and outgoing long wave radiation there is little variation in the location on the Taylor diagram (Fig. 6(a) & (b)). For SLP patterns the scatter does not appear much greater than would be expected by chance for both HadAM3 and HadAM3P. Precipitation is generally slightly worse for the HadAM3 optimised cases than the standard configuration with spread to smaller correlations and larger RMS differences. For the HadAM3P configuration the optimisation slightly improves the spatial patterns of precipitation (Fig. 6(a)). For RSR pattern correlations

and centred RMS differences show the largest spread across the variables we consider with some of the 7-parameter optimised cases an improvement on the standard configuration. For HadAM3P the centred RSR patterns are worse than the standard HadAM3P case. The optimised HadAM3 cases for Relative Humidity at 500 hPa scatter around the standard cases with some better and some worse though as other variables the differences are small. Overall, the HadAM3 optimised and standard values are very similar.

## 30  3.3   Coupled Model results

To test if calibrating atmospheric parameters results in reasonable coupled models we took the calibrated parameters from all successful 7- and 14-parameter cases in a set of control simulations off HadCM3. The surface temperature adjusts in the first decade (Fig. 7(a)) though the deep ocean is still adjusting during the 20 year simulations (Fig. 7(b)). Williamson et al. (2013) estimated that pre-industrial temperatures were 13.6°C with a robust error estimate of ±0.5°C. We claim that a coupled model



is "good" if the global and time average of its surface air temperature for years 10-19 is consistent with that estimate. The standard configuration is, just, within this range and as noted by Gordon et al. (2000) HadCM3 is somewhat too cool.

For the HadAM3-7 cases we find eight of the parameter combinations produce temperatures within the target range (Table 5). For the HadAM3-14 cases five out of seven parameter combinations give temperatures within the target range. All four cases that fail are too cold rather than too warm. As we start from the standard configuration we may be more able, in the 20 year simulations we did, to identify cooling rather than warming biases. Though all atmospheric models were constrained to be in rough energy balance the individual fluxes are less constrained. For three of the cases that cooled RSR rapidly increases over the first 5 years with OLR decreasing over the same period. However, the RSR increases by more than the OLR decreases so the coupled model is out of balance and cools (Fig. 7). This may be due to negative cloud feedbacks in these model configurations. The remaining coupled models show a range of OLR and RSR values but are generally stable.

We now examine if there is any relationship between properties in the atmospheric model simulation and the coupled model simulation. Above we showed that RSR changes were somewhat larger than OLR changes and, across the optimised parameter sets, RSR variability was larger, relative to its uncertainty, than OLR variability (Fig. 3). Thus, we focus on relationships between global-average RSR and various properties of the coupled models. We examine the 10 year global-average for 2001-2010 from the independent atmospheric simulations and years 10-19 from the Control simulations.

For surface air temperature and volume average ocean temperature there is a relationship between atmospheric model RSR and Control values with an increase in atmospheric RSR leading to cooling in the coupled model (Fig. 8) though with some scatter around this general relationship. Uncertainties on the regression are small. We also find an inverse relationship between the strength of the Atlantic Meridional Overturning Circulation (AMOC) in the Control simulation and the Atmospheric RSR. Likely because cold models have a stronger AMOC. Similar results hold true for NH Snow area and Sea Ice Area. For Land Precipitation the scatter is too large to conclude there is a strong linear relationship. We repeated this analysis using OLR from the atmospheric simulations and found similar, though opposite signed and weaker, results. This likely arises from the constraint on the net flux meaning that enhanced RSR must be balanced by reduced OLR. Note that the range of atmospheric RSR values is within the estimated uncertainty estimate for RSR (See Fig. 1 of T13) and so all cases (after running the atmospheric optimisation) are "good".

.

# 4 Discussion

Our results suggest that calibrating the atmospheric component of a coupled models to multiple observations is computationally feasible with the resulting coupled models behaving well much, but not all, of the time. However, we found that calibration of 14 parameters was less successful than that of 7 parameters. We now investigate potential reasons for this by looking at the Jacobian matrices from all 7- and, non-random, 14-parameter studies. We also examine the Jacobian of the HadAM3P-7 parameter cases to see if changing resolution affects the Jacobian and so might explain the failure of the HadAM3P-13r6 case.



We computed Jacobians for each iteration with the parameters normalised by their range so 0 (1) is the minimum (maximum) value and normalised each bias by its simulated internal variability. To see which parameters have strongest effect on simulated observations we compute the mean, over all iterations, of the *absolute* Jacobian values. We compare this to the changes that internal variability by comparison with a folded Normal distribution (Leone et al., 1961) using a 90% critical value. To derive

the parameters for this distribution we assume that the underlying normal distribution arises from the difference of two random distributions with unit variance and zero mean ($\sigma = \sqrt{2}, \mu = 0$).

We see that in the 7-parameter cases (Fig. 9(a)) that all parameters, except ICE_SIZE, have an significant impact on Net Flux and the cost function ($F$). ICE_SIZE affects both OLR and RSR outside the NH extra-tropics but these must offset one another leading to a small impact on Net Flux and on $F$. All parameters affect RSR in the tropics and almost all affect it in the

extra-tropics. In contrast, tropical OLR is significantly affected by only three parameters (ICE_SIZE, VF1 & ENTCOEFF) with the remaining four parameters having little impact on OLR. NH Extra-tropical Land Precipitation & SLP and Tropical SLP are not significantly affected by any of the parameter perturbations. In the SH land temperature is only weakly affected by changes in VF1 while Precipitation is not significantly impacted. These likely reflect the small land area in the SH and the resulting increase in internal variability. In terms of relative importance we see that changes in ENTCOEFF parameter has the

most impact on the cost function with RSR being most effected by parameter changes. In contrast we see that ICE_SIZE has least impact on the cost function and extra-tropical Land Precipitation and Pressure gradients being unaffected by parameter perturbations.

Examining the 14 parameter Jacobians (Fig. 9(b)) we see that four of the additional seven parameters have a significant impact on the cost function. However, of these only DYNDIFF has a more than small impact on the cost function. For these 6

parameters our preliminary tuning (see above) had led to parameter perturbations that were large relative to the range (Table 1). As with the seven parameter cases ENTCOEFF has the largest impact with RHCRIT the second most important. However, from this larger set of parameters all simulated observations, except Southern Hemisphere Land temperature and precipitation are effected. The CHARNOCK, ICE_SIZE and ALPHAM parameters have no significant impact on the cost function. Further the CHARNOCK parameter was perturbed by about one-third of its range meaning little freedom to further perturb it.

The mean of the absolute Jacobians between the 14- and 7-parameter cases shows some differences in detail (compare Fig. 9(a) with (b)) suggesting that the Jacobians are, as expected, not constant. More detailed examination of this (not shown) suggests that within an individual study, after the first iteration, the Jacobians are fairly stable but within different parts of parameter space the Jacobians differ *even* if the final states appear quite similar.

Looking at the absolute Jacobians from the HadAM3-7 computations ( Fig. 9(c)) we see differences from the two HadAM3

results with VF1 and RHCRIT no longer having a significant impact on the cost function. This likely arises from the smaller impacts on the Net Flux, than in HadAM3, which has, in our constrained optimisation, a large effect on the cost function. In contrast the effect of ENTCOEFF and CT on the cost function is much larger in HadAM3P than it is in HadAM3.

Regarding the poor performance of the HadAM3-14r8 algorithm, it is unclear at this stage precisely what has caused it, given that HadAM3-14r6 behaves very well. We speculate that this may be caused by noise contamination, and that the less

parameters we perturb in the algorithm, the smaller the chance of seeing the effect of noise. Alternatively there could be



instability in the randomised algorithmic variant, again due to noise. We note that if the cost function is smooth and accurate derivatives were available, one can easily observe improving rates of convergence for randomised block Gauss-Newton variants the more parameters one chooses in the block (Eizenberg, 2015).

As part of the development of our approach we carried out four trial cases where we started from parameter sets (Table 3) with the largest climate sensitivities. We present results from them to explore the senitivity of our results to changes in the algorithm and cost function. The cases are:

**trial7#diag** No differences except for starting parameter values.

**trial7#cons** Reduce $\mu$ from 0.01 to 0.001.

**trail7#17obs** Use $\alpha$ values of 1, 0.7 & 0.3, increase $\mu$ to 1, remove LAT from observations used, use a non-diagonal covariance matrix for observational error and so have a very different cost function.

**trial7#15m** Run model for 15 months, compare model and observations from March 2000 to February 2001, and scale internal covariance matrix by 5.

All trials (Fig. 10) converged to states with cost functions similar, though slightly larger, than the reference model. Independent simulations have cost functions slightly larger than the cases from the optimisation with the difference being largest for trial7#15m. However, all cases produced models that cooled and had temperatures outside the range of acceptable coupled models. This suggests that the Gauss-Newton algorithm converges for a range of cost functions but not necessarily to a case that produces an acceptable coupled model.

Zhang et al. (2015) reported successful optimisation of IAP LASG version 2 atmospheric model. They focused on only seven parameters and, unlike us, used a root-mean-square error between simulation and observations normalised by the standard deviation of the standard simulation. They considered a broader range of variables than we did though used the older ERBE data rather than the recent CERES data. Unlike us they screened out three of the parameters using the Morris (1991) method. Starting from the default parameter set they improved their skill score by a small amount thought, unlike us, did not test if this was a selection effect. Their best algorithm took about 60 iterations broadly consistently with out expected number of about 70 iterations.

Various other studies have attempted to produce stable coupled models. Yamazaki et al. (2013) used emulation to find parameter sets that would be expected to produce, in HadaCM3, RSR and OLR values that, relative to the standard configuration of HadCM3, are within the uncertainty limit of Tett et al. (2013c). They found global average temperatures of $289.9 \pm 3.6$ K which is a range larger than that of the CMIP3 and CMIP5 ensembles. The uncertainty estimate used in their study includes several sources of uncertainty in addition to observational error. Restricting their analysis to model configurations that have RSR and OLR values within 20% of that uncertainty range, which has a net TOA flux range of $\pm$ 1.1 W m$^{-2}$, they found those models had a broad range of climate sensitivities and a global mean temperature range of 286-291K (Fig. 5 of Yamazaki et al. (2013)).



Irvine et al. (2013) used a latin-hypercube design to produce 200 versions of the coupled atmosphere-ocean HadCM3 model with 8 parameters being perturbed. They ran each version for 20 years, estimated the final equilibrium temperature and discarded cases which were outside the range $13.6 \pm 2°C$. From their 200 initial versions they found 20 cases that met that criteria. How does the computational cost of this compare with our approach of perturbing the atmospheric model then coupling the

perturbed atmosphere to the ocean model? The nearest cases we have are the HadAM3-7 cases which need an expected 68 evaluations (Table 5) each of 6.25 years for a total of 425 years of atmospheric simulation. As the atmospheric model is about half the cost of the coupled model this is equivalent to about 210 years of coupled simulation. We then need to carry out 10/8 coupled model simulations each of 20 years to get one that is within observational uncertainty for a grand total of 225 coupled-model years. This is approximately the same computational resource that Irvine et al. (2013) need but produces coupled models

in better agreement with the pre-industrial temperature estimates.

## 5    Conclusions

We have shown it is possible to automatically calibrate HadAM3 and HadAM3P and, much of the time, ending up with models that appear similar, or for HadAM3P, better than the standard configuration. We used two variants of the Gauss-Newton algorithm. One in which all parameters were varied and a second random block-coordinate variant in which a sub-set of the

parameters, chosen at random on each iteration, were varied. For the studies in which we perturbed 7 parameters in HadAM3 we found that all cases converged taking an average number of 68 evaluations for a total of 425 simulated years.

For the 14-parameter cases we used both the standard Gauss-Newton algorithm and a variant where a random number of parameters were selected. We tried two random cases. One in which 6 parameters were perturbed and an another in which 8 were perturbed. For each algorithm five studies starting from the same initial parameter choices were carried out. We find large

differences in the performance of these algorithms with the 6 random perturbation algorithm performing best and the 8 random perturbation cases worst with the standard Gauss-Newton algorithm performing intermediately. The 6-random case needs an expected number of 82 evaluations (or 512 simulated years) and, on average, produces models that are slightly better than the standard configuration. Given the sensitivity of the total iterations needed to produce acceptable models to small changes in the number of random parameters then further work is needed to determine how many parameters should be perturbed.

As discussed above, the poor performance on the 14 parameter case seems to be due to some of the parameter perturbations having only a small impact on the cost function leading to noise contamination of the line-search vector causing the algorithm to head in random directions. The poor performance of the random variant that perturbed 8 out of the 14 parameters at random may also be due to noise contamination arising again from unimportant parameters being included similarly to the full 14 parameter case or causing some kind of algorithm instability. We recall that Eizenberg (2015) illustrated numerically that for

smooth problems with available derivatives, the randomised variants' rates of convergence improve continuously the more parameters are included in the blocks being perturbed.

We also found several different parameter combinations led to models that were broadly comparable with the standard configurations. This suggests that HadAM3 exhibits equifinality (Beven and Freer, 2001) with different parameter sets leading



to models that appear similar. Further, many, though not all, of the resulting coupled models are consistent with pre-industrial temperatures without any need for flux correction. This is a significant advance on previous work using perturbed physics models which have generally had to flux correct the resulting models.

If these techniques could be successfully applied to state-of-the-art models it would be practical to:

5    – generate perturbed models to test if an observationally constrained ensemble has a narrow range of climate feedbacks.

– add new parametrisations of processes to a model then recalibrate the model.

– explore the effect of changing resolution without large changes in the simulation of large scale climate.

Though our algorithm works reasonably well for a modest number of parameters it would benefit from a better understanding of the effect of noise on it. Both the line-search through a selection effect and the computation of the Jacobian/Hermitian 10  matrices are effected by noise. A better algorithm would identify parameters that did not appear to impact the cost function and remove them from the analysis as done by Zhang et al. (2015). Another potential approach might be to update the components of the Jacobian from the previous iterations values depending on the relative amount of noise contamination in them. We hope that our derivative-based experience with randomised block-variants of Gauss-Newton – where the rates improve the larger the size of the block of parameters being perturbed – would then be observed here as well. This would further allow us to 15  leverage /quantify the trade-off's of the lower evaluation cost per iteration of the small-block randomised variants against their respective global rates of convergence.

Our work focused on optimisation rather than the cost function. We used a cost function based on crude estimates of observational uncertainty and a subjective choice of large scale observations. Future work would benefit from much better estimates of observational uncertainty and an objective means of selecting observations. One approach might be to chose 20  observations which we have good evidence matter for climate feedbacks or other properties of the model we are concerned about.

Nevertheless, our results suggest that it is possible and computationally feasible to automatically calibrate the atmospheric component of a climate model and generate a plausible coupled model.

## 6   Code and data availability

25  All data and software are available from CEDA at http://catalogue.ceda.ac.uk/uuid/889fc3c877e8447bb7b2a100ef17a3f4.

We implemented and developed the algorithms described above using bash shell scripts and ipython (Pérez and Granger, 2007) with the numpy (Van Der Walt et al., 2011), pandas(http://pandas.pydata.org/) and iris (http://scitools.org.uk/iris/docs/latest/index.html) modules. Each iteration was managed using Grid Engine with runs of the climate models each being followed by a job that computed the simulated observables. A final job in each iteration tested for termination and, if required, set up 30  the next iteration. Visualisation was done using Matplotlib (Hunter, 2007) supplemented with seaborn(https://stanford.edu/~mwaskom/software/seaborn/).



## Appendix A: Glossary

$p$ vector of parameter values.

$p_k$ vector of parameter values at iteration $k$.

$O$ vector of target observations.

$N$ Number of observations.

$P$ Number of perturbed parameters used in Gauss-Newton algorithm.

$P_{\mathrm{rand}}$ Number of parameters, selected at random, that are perturbed, in each iteration, in random block coordinate variant of Gauss-Newton algorithm.

$S$ Vector of simulated observations.

$S_k$ Vector of simulated observations at iteration $k$.

$E_c$ Mean evaluation for cases that converged.

$E_f$ Mean evaluation for cases that filed to converge.

$f$ Fraction of studies that failed.

$F(p)$ Cost-function being minimized.

$C_i$ Covariance matrix representing internal variability.

$C_O$ Covariance matrix representing observational error.

$C$ Covariance matrix representing combined internal variability and observational error.

$k$ Iteration number.

$\mathcal{J}$ Jacobian of matrix with $\mathcal{J}_{ij} = \frac{\partial S_i}{\partial p_j}$.

$\mathcal{H}$ Hessian ($\mathcal{J}^T C^{-1} \mathcal{J}$).

$J$ Finite difference estimate of Jacobian.

$H$ Finite difference estimate of Hessian ($J^T C^{-1} J$).

$\mu$ Weight on constrained optimisation.

$\mathbf{N}$ Number of studies in case.



**N**<sub></sub>**Atmos**  Number of converged cases with cost function similar to standard model.

**N**<sub></sub>**Control**  Number of Coupled Control simulations consistent with pre-industrial temperatures (Williamson et al., 2014)

*Author contributions.* Tett, Cartis & Mineter concieved the study. Mineter implemented the software framework. Yamazaki implemented the GN algorithm, with guidance from Cartis, within the framework, carried out the 7 parameter studies and did preliminary analysis. Eizenberg, supervised by Cartis, implemented and tested the random variant of Gauss-Newton algorithm. Tett re-engineered framework, carried out 14 parameter and high resolution cases. Tett wrote paper and all commented upon it.

*Acknowledgements.* This work was funded by NERC (NE/L012146/1) with simulations and post-processing done on the Edinburgh Compute and Data Facility. We thank Dan Williamson (Exeter) for providing R-code to compute parameters from meta-parameters and Sam Pepler (BADC) for assistance in archiving data and software.



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





**Table 1.** Parameters, default values and allowed ranges and perturbations. Shown for each parameter name are the component of HadAM3 they are from, the default value, allowed range, perturbations used in HadAM3-7 cases ($\Delta_1$), and perturbations used all HadAM3-14 cases and HadAM3P cases ($\Delta_2$). For more information on the parameters see Yamazaki et al. (2013). Where HadAM3P values or ranges differ from HadAM3 these are shown in brackets.

| Parameter | Component | Default Value | Range | $\Delta_1$ | $\Delta_2$ | |
|---|---|---|---|---|---|---|
| VF1 | Cloud | 1(2) | 0.5–2 | 0.5 | 0.1 | ms$^{-1}$ |
| RHCRIT[b] | Cloud | 0.7 | 0.6–0.9 | 0.05 | 0.01 | |
| ICE_SIZE | Radiation | 30 | 25–40 | 1.5 | 1.5 | $\times10^{-6}$ m |
| ENTCOEFF | Convection | 3 | 0.6–9 | 0.6 | 0.15 | |
| EACF[b] | Cloud | 0.5 | 0.5–0.7 | 0.02 | 0.02 | |
| CT | Cloud | 10(60) | 5–40(5–100) | 2 | 1 | $\times10^{-5}$ s$^{-1}$ |
| CW_LAND [a] | Cloud | 2(10) | 1–20 | 1 | 2 | $\times10^{-4}$ kg m$^{-3}$ |
| DYNDIFF[d] | Dynamics | 12 [e] | 6–24 | - | 2 | hours |
| KAY_GWAVE[f] | Dynamics | 20(18) | 10–20 | - | 4 | $\times10^{3}$ |
| ASYM_LAMBDA | Boundary Layer | 0.15 | 0.05–0.5 | - | 0.15 | |
| CHARNOCK | Boundary Layer | 12(10) | 12–20(10–20) | - | 3 | $\times10^{-3}$ |
| G0 | Boundary Layer | 10 | 5–20 | - | 4 | |
| Z0FSEA | Boundary Layer | 13 | 2–50 | - | 20 | $\times10^{-4}$ m |
| ALPHAM[c] | Radiation | 0.5 | 0.5–0.65 | - | 0.06 | |

[a] controls CW_SEA; [b] these parameters set values on all model vertical levels; [c] also sets DTICE,[d] sets diffusion values on all levels for temperature and humidity;[e] for HadAM3P the default value is 3 (11) hours for temperature (humidity) with $\nabla^2$ diffusion for temperature and humidity throughout the atmosphere; [e] also controls KAY_LEE_GWAVE.

**Table 2.** Target values for optimisation cases. Each row corresponds to a region. The target value for net flux into the Earth is 0.5 Wm$^{-2}$.

| | OLR (Wm$^{-2}$) | RSR (Wm$^{-2}$) | LAT (K) | LP (mm/day) | $\Delta$ SLP (hPa) | T500 (K) | q500 (%) |
|---|---|---|---|---|---|---|---|
| NHX | 223.0 | 102.3 | 275.8 | 1.44 | 3.31 | 251.4 | 53.4 |
| Tropics | 259.9 | 94.2 | 297.6 | 3.12 | 1.79 | 266.7 | 33.9 |
| SHX | 216.1 | 108.1 | 287.4 | 1.93 | – | 248.9 | 52.7 |





**Table 3.** Normalised initial parameters for 7- & 14-parameter, HadAM3P and trial cases. All parameters are normalised by their expert based ranges with 0 (1) being the minimum (maximum) values. Values not shown use the default HadAM3 (or HadAM3P) values. Parameter names are shortened to their first three characters. Initial parameters from the two 14-parameter random Gauss-Newton algorithms are not shown as they match the equivalent values from the standard Gauss-Newton algorithms. Similarly only the HadAM3P13r6 and trial7#diag cases are shown as other HadAM3P and trial7 cases use the same values.

|  | VF1 | RHC | ICE | ENT | EAC | CT | CW | DYN | KAY | ASY | CHA | G0 | Z0F | ALP |
|---|---|---|---|---|---|---|---|---|---|---|---|---|---|---|
| HadAM3-7#03 | 0 | 1 | 1 | 0 | 1 | 1 | 0 | – | – | – | – | – | – | – |
| HadAM3-7#04 | 1 | 1 | 1 | 1 | 0 | 1 | 0 | – | – | – | – | – | – | – |
| HadAM3-7#05 | 1 | 0 | 1 | 0 | 0 | 1 | 1 | – | – | – | – | – | – | – |
| HadAM3-7#06 | 1 | 0 | 0 | 0 | 0 | 0 | 0 | – | – | – | – | – | – | – |
| HadAM3-7#07 | 0 | 0 | 1 | 0 | 0 | 0 | 0 | – | – | – | – | – | – | – |
| HadAM3-7#08 | 1 | 0 | 0 | 0 | 1 | 1 | 0 | – | – | – | – | – | – | – |
| HadAM3-7#09 | 0 | 0 | 0 | 1 | 1 | 1 | 0 | – | – | – | – | – | – | – |
| HadAM3-7#10 | 1 | 0 | 0 | 1 | 1 | 0 | 1 | – | – | – | – | – | – | – |
| HadAM3-7#11 | 1 | 1 | 1 | 0 | 0 | 1 | 1 | – | – | – | – | – | – | – |
| HadAM3-7#12 | 0 | 1 | 1 | 1 | 0 | 0 | 0 | – | – | – | – | – | – | – |
| HadAM3-14#1 | 1 | 0 | 0 | 1 | 0 | 1 | 1 | 1 | 1 | 0 | 1 | 0 | 0 | 0 |
| HadAM3-14#2 | 0 | 0 | 1 | 0 | 1 | 1 | 0 | 0 | 0 | 1 | 0 | 0 | 0 | 0 |
| HadAM3-14#3 | 0 | 1 | 0 | 0 | 1 | 0 | 1 | 0 | 0 | 0 | 1 | 1 | 1 | 1 |
| HadAM3-14#4 | 1 | 0 | 1 | 1 | 1 | 1 | 0 | 0 | 0 | 0 | 1 | 1 | 0 | 1 |
| HadAM3-14#5 | 0 | 0 | 1 | 0 | 0 | 0 | 0 | 0 | 0 | 0 | 1 | 0 | 0 | 1 |
| HadAM3P-13r6 | 0.33 | 0.33 | 0.33 | 0.29 | 0 | 0.05 | 0.05 | – | 1 | 0.22 | 0.20 | 0.33 | 0.23 | 0 |
| trial7#diag | 0 | 1 | 1 | 0 | 0 | 0.14 | 0.05 | – | – | – | – | – | – | – |

**Table 4.** Count of unique $\alpha$ in 7-parameter HadAM3 optimisations.

| $\alpha$ | 0.01 | 0.10 | 0.30 | 0.70 | 1.00 |
|---|---|---|---|---|---|
| Count | 11 | 12 | 9 | 9 | 17 |



**Table 5.** Algorithm Summary. For each algorithm is shown the expected number of iterations, evaluations, mean cost from final optimisation simulations ($F$), mean cost from independent atmospheric simulations ($F_i$), mean cost from standard configuration ($F_s$), number of cases ran (N), number that converged ($N_{Atmos}$) and number that were "good" coupled models ($N_{Control}$). The algorithm name shows the model, number of parameters varied and if the random block-coordinate variant is used then $rP_{rand}$ denotes that $P_{rand}$ parameters, selected at random on each iteration, were perturbed.

| | Iterations | Evaluations | $F$ | $F_I$ | $F_S$ | N | $N_{Atmos}$ | $N_{Control}$ |
|---|---|---|---|---|---|---|---|---|
| HadAM3-7 | 5.60 | 68.20 | 4.77 | 4.97 | 5 | 10 | 10 | 8 |
| HadAM3-14 | 8 | 138.50 | 4.86 | 5.21 | 4.68 | 5 | 2 | 1 |
| HadAM3-14r6 | 9 | 82.25 | 4.18 | 4.64 | 4.68 | 5 | 4 | 3 |
| HadAM3-14r8 | 21 | 236 | 4.90 | 5.46 | 4.68 | 5 | 1 | 1 |
| HadAM3P-13r6 | — | — | — | — | 7.57 | 1 | 0 | 0 |
| HadAM3P-7 | 3 | 31 | 7.52 | 7.64 | 7.57 | 1 | 1 | 0 |
| HadAM3P-7r3 | 5 | 31 | 6.17 | 6.15 | 7.57 | 1 | 1 | 0 |





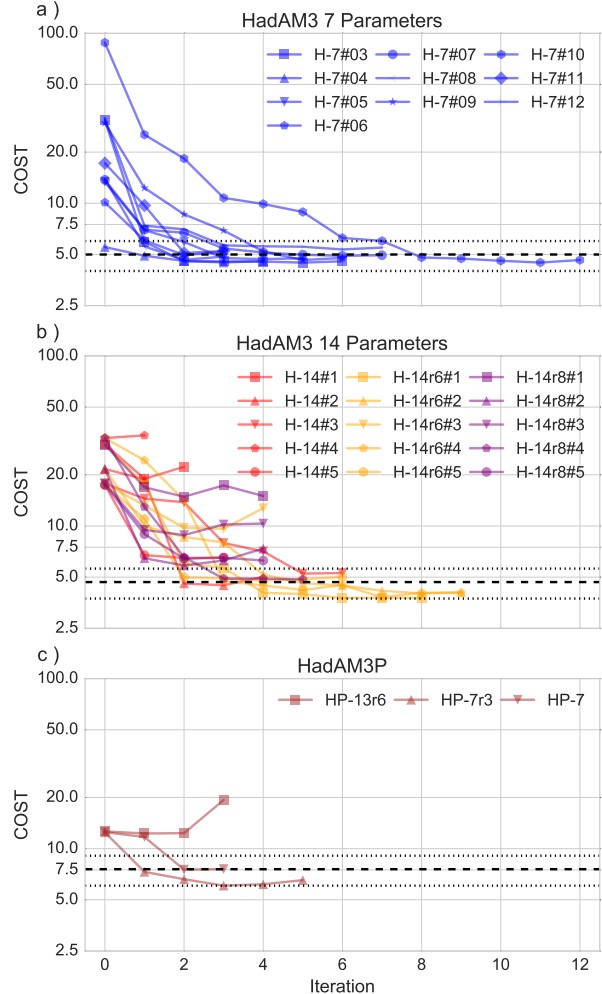

**Figure 1.** Minimum cost function for "line-search" component of algorithm (y-axis) versus iteration number (x-axis) for HadAM3 7-parameter (a), HadAM3 14-parameter (b) and HadAM3P (c) optimisations. Iteration 0 is the cost function for the initial parameter values. The y-axis uses a logarithmic scale. In each subplot the dashed black line show the cost function for the standard model while the dotted lines show the ±20% ranges on this showing perturbed models comparable with the standard configuration. Coloured lines and symbols use the key shown in each sub-plot. For plots b) & c) labels with $rP_{rand}$ had $P_{rand}$ random parameters perturbed on each iteration.





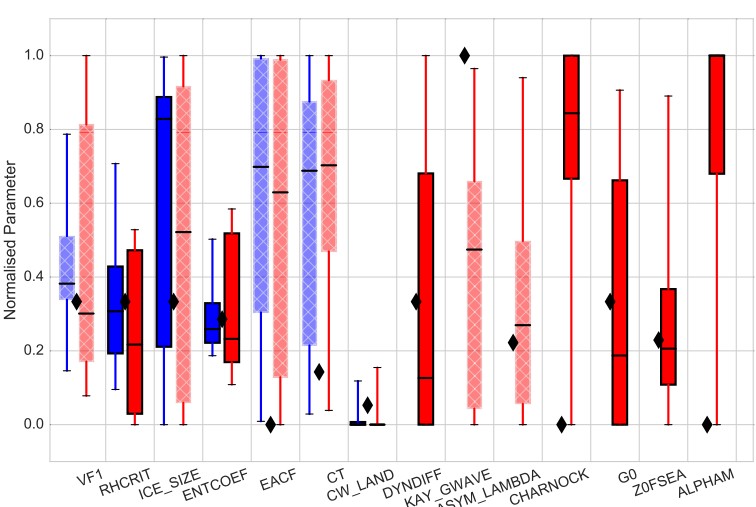

**Figure 2.** Normalised parameter values (y-axis) for default HadAM3 (Black diamond), optimised 7-parameter (blue) and 14-parameter (red) cases. x-axis shows parameter. Shown are the 25-75 % values (boxes), the median values (black horizontal line) and the ranges (vertical lines). Pale hatching shows where the null hypothesis that the optimised parameters are uniformly distributed in the range 0-1 is not rejected at the 10% value using a Kolmogorov-Smirnov test. Where this null hypothesis is rejected the inter-quartile boxes are edged in black.



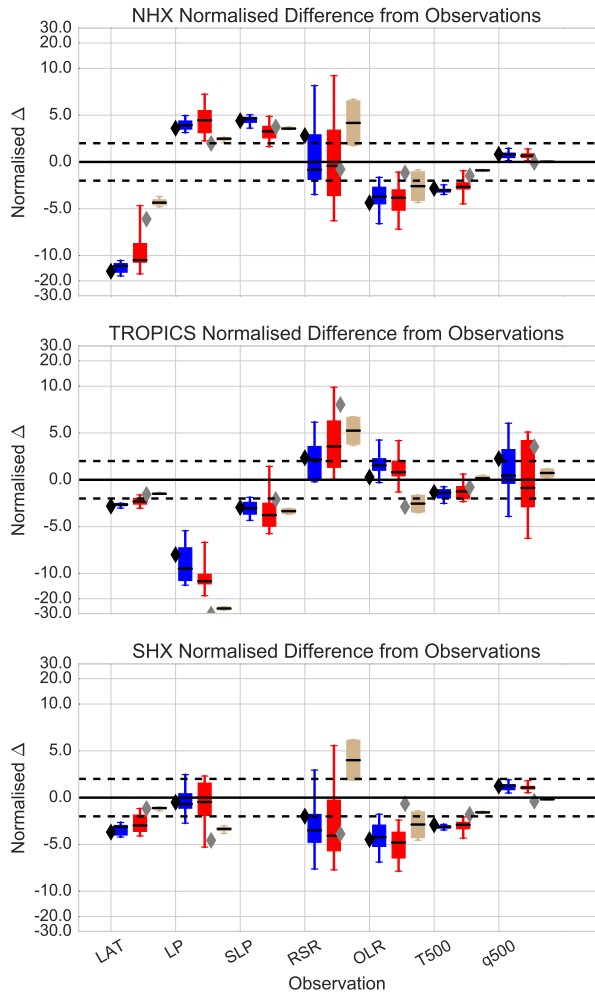

**Figure 3.** Normalised simulated minus observed distributions (y-axis) for 7 (blue) and 14-parameter (red) cases. Top panel is northern hemisphere extra-tropics, middle tropics and lower southern hemisphere extra-tropics. Boxes and whiskers as Fig. 2, with observations on x-axis. Also shown are the optimised HadAM3P cases (tan boxes/lines) and the standard HadAM3 (black diamonds) and HadAM3P (grey diamonds) values. All differences are normalised by the square root of the diagonal elements of $C$. The dashed lines show $\pm 2$. The scale is linear between 5 and -5 and logarithmic outside that range.



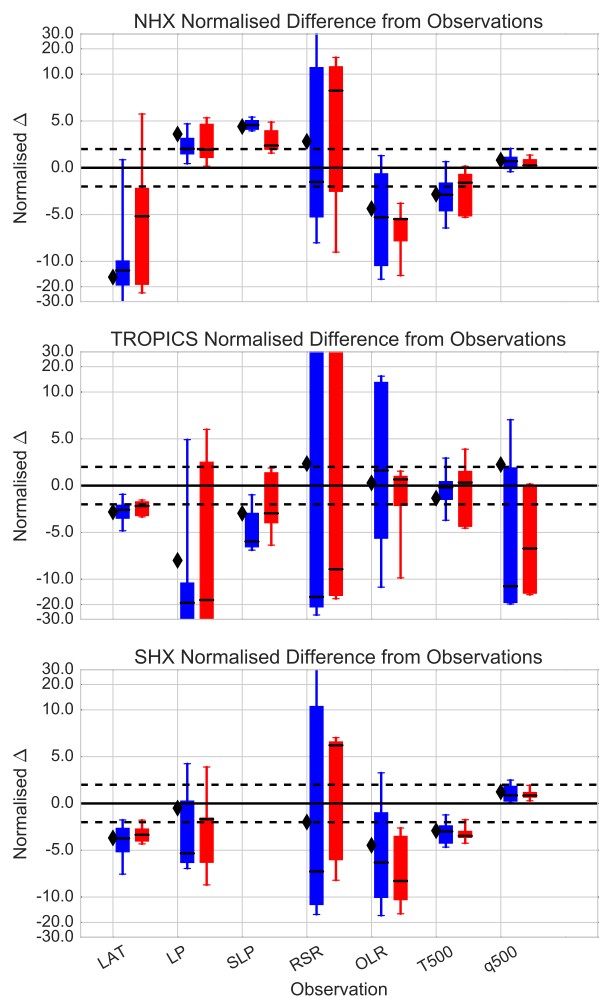

**Figure 4.** As Fig. 3 except shown are initial extreme random parameter choices for 7-(red) and 14-parameter (blue) cases.





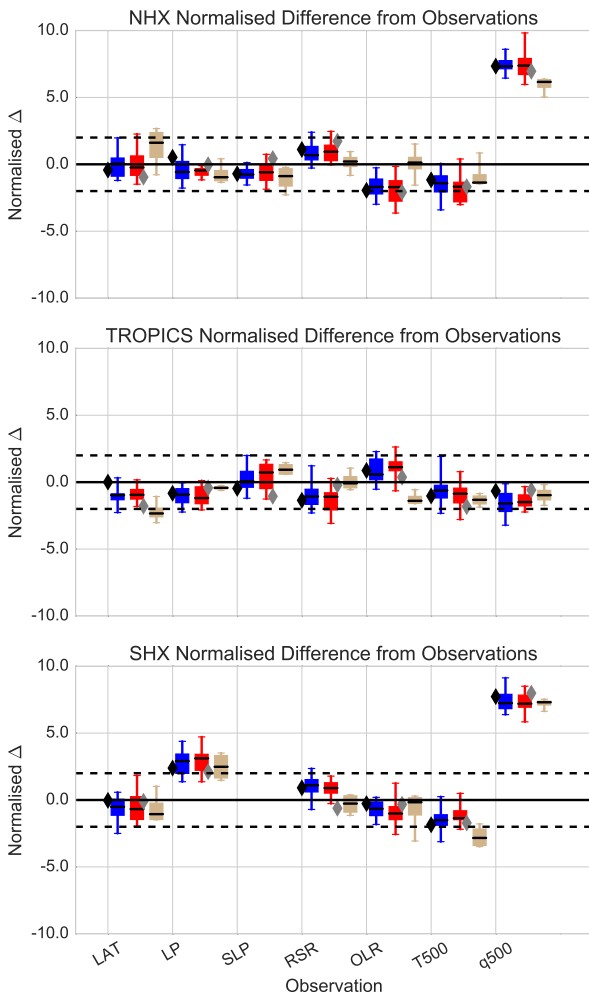

**Figure 5.** As Fig. 3 except shown are differences between simulated, from 2-member ensemble, and observed change between 2005-2010 and 2000-2005. All values are standardised by $3\sigma$ internal variability computed from the 100 member 2000-2005 HadAM3 ensemble.





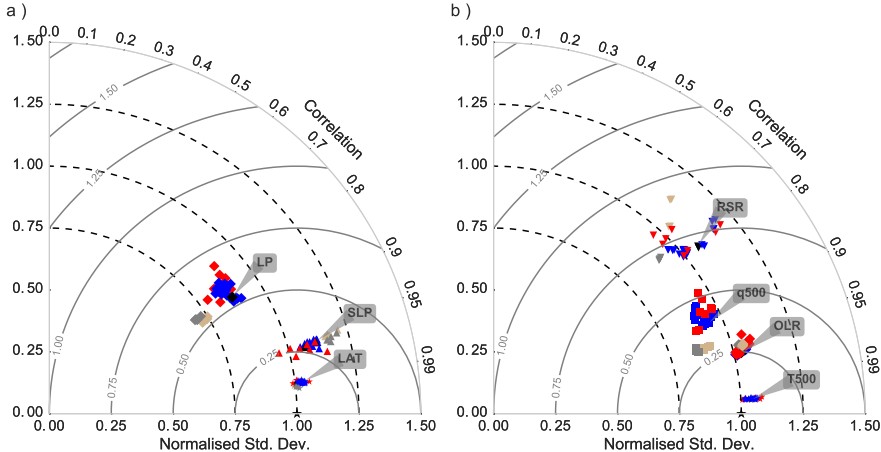

**Figure 6.** Normalised Taylor Diagrams for Land Air Temperature (asterisks), Mean Sea Level Pressure (triangles), Land Precipitation (diamonds) (a) and Temperature at 500 Hpa (asterisks), OLR (diamonds), Relative Humidity at 500 hPa (squares) and Reflected Shortwave Radiation (inverted diamonds) (b). Default HadAM3 (HadAM3P) values are gray (black). Independent 7-parameter and 14-parameter cases are shown in blue and red respectively. The labels point to the standard HadAM3 values.

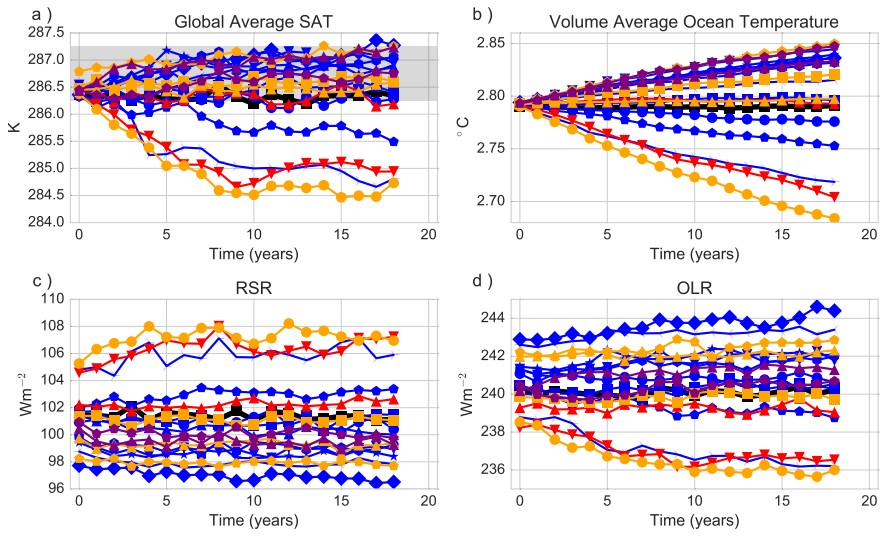

**Figure 7.** Timeseries, from Control ocean-atmosphere simulations, of annual-average global-average: 1.5m temperature (a), Mean volume Average Ocean Temperature (b), RSR (c) and OLR (d). Colours and markers as Fig. 1 with Standard configuration shown as thick black line. Gray shading on (a) shows uncertainty range for pre-industrial global-average air temperatures from Williamson et al. (2014)





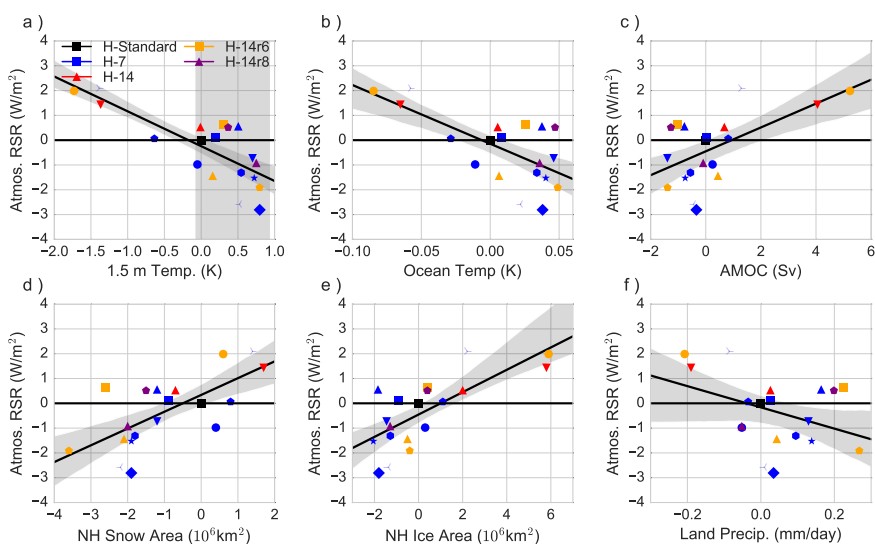

**Figure 8.** Scatter plot (symbols as Fig. 1; colours in key) of optimal minus standard configurations for 2000-2010 mean RSR from atmospheric only simulations against the coupled Control mean for years 10-19 of: 1.5 m temperature(a), volume average ocean temperature(b), coupled Atlantic Meridional Overturning Circulation(c), Northern Hemisphere area where snow mass is > 6kg m$^2$(d), Northern Hemisphere ice area(e) and Land precipitation(f). Black lines and grey regions show best fit regression line and 90% confidence interval on the regression lines. Vertical grey region in sub-plot (a) show the difference between pre-industrial global-average air temperatures and the standard configuration; configurations in this region are "good".





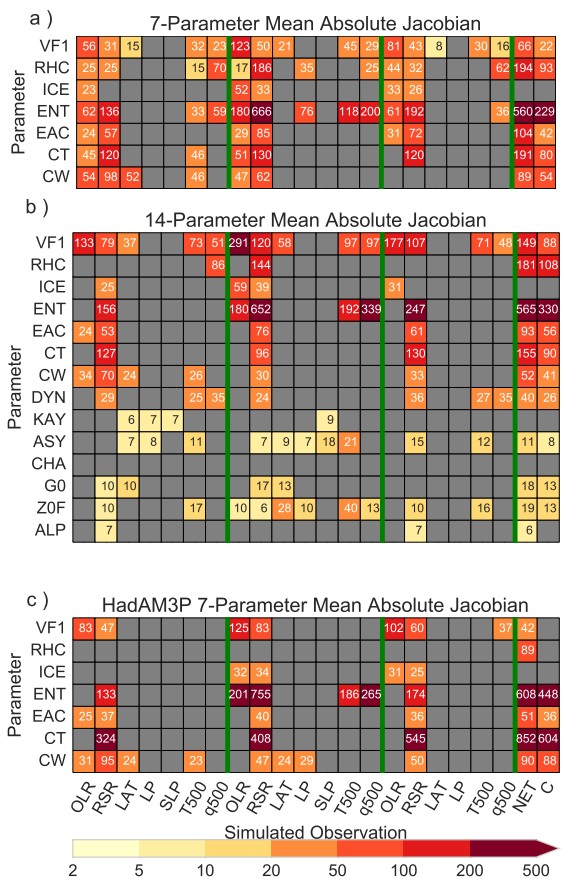

**Figure 9.** Mean of absolute Jacobian for 7- (a), 14- (b), and HadAM3P 7- (c) parameter cases. All values are normalised by parameter range and internal variability (see main text). Colours change logarithmically using scale at bottom of plot. Gray regions are where absolute Jacobian, for one evaluation, is not significantly different at 90% level from zero (See main text). Y-axis shows parameter names shortened to first three characters while x-axis on bottom plot shows simulated observations with region deleted. Observations are organised into three groups of seven as NH extra-tropics, tropics and SH extra-tropics split by green vertical lines. Also shown, on the right, are the Net Flux (Net) and Cost (C).





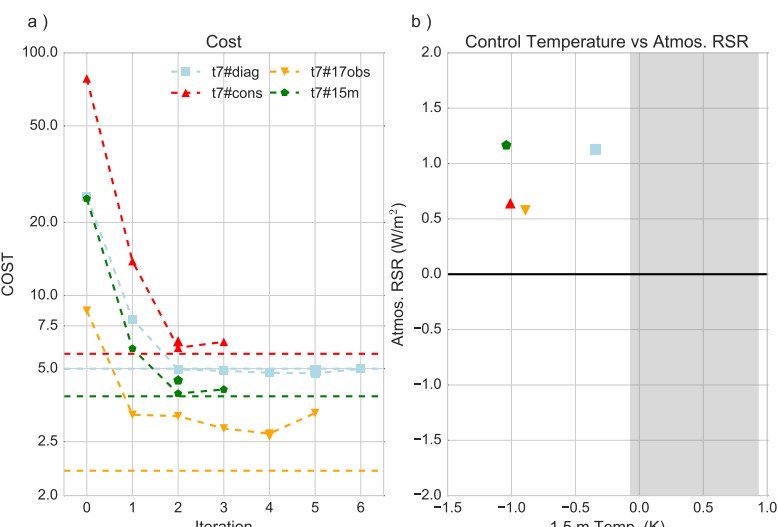

**Figure 10.** a) Cost as function of iteration for four trial 7-parameter cases (lines and symbols). Also shown are the cost functions for the standard model for each case (coloured horizontal lines) with the colour corresponding to the trial case. Large symbol for each trial case show cost from independent atmospheric simulation with other details as Fig. 1. b) Scatter plot (symbols as a) of optimal minus standard configurations for 2000-2010 mean RSR from atmospheric only simulations against the coupled Control mean for years 10-19 of 1.5 m temperature. Vertical grey region show estimates of the difference between pre-industrial global-average air temperatures and the standard configuration; configurations in this region are "good".