# Peer review of "Calibrating Climate Models Using Inverse Methods: Case studies with HadAM3, HadAM3P and HadCM3"

_Geoscientific Model Development, 2016_

## Referee Comment (RC1) · Anonymous Referee #1 · 15 Mar 2017

GENERAL COMMENTS The manuscript discusses using Gauss-Newton line-search algorithm in optimising/tuning the atmospheric and coupled versions of Hadley Centre model. The optimisation uses a multi-criteria target, that includes 5 additional constraints to the authors' previous work. Optimisation is done for HadAM3-model by perturbing 7-/14-parameters. Similarly, parameter optimisation for HadAM3P-model is done by perturbing 7-/13-parameters. Additionally, tests are done for only perturbing a subset of the parameters at once. The optimisation seems to work well for the 7-parameter cases, but less so when the number of simultaneous parameter perturbations is increased. The results show the the GN line-search algorithm is able to optimise the parameter values in the training set, but also produces atmospheric models that can be used in coupled climate runs.

The manuscript is well written, and the results are well presented and interesting. The authors have done a good job in presenting numerous aspects of both the optimisation process as well as how the optimised parameters impact the model in climate runs. I have quite a few comments, but they probably won't require too much work. I am suggesting a "minor revision" for the manuscript.

SPECIFIC COMMENTS Eq.3: Could you elaborate on "eˆi is the ith coordinate vector", the Nocedal and Wright book is not readily available.

P5 L15: "...perturb towards the middle...". Perturbing always towards the centre is most likely a good idea, but would the algorithm converge if the optimal parameter values were at the edges of the "allowed" space? Just thinking about hypothetical cases where the default parameter values would be ill set. (Just interested in your thoughts about this, not necessary to add any text.)

P5 L22: "several steps", how is this defined? Is the number of steps dynamic or static? Linked to this, could you make sure that you include the total number of steps also when you talk about how many iterations it took for the algorithm to converge/stop (e.g. P10 L3).

Eq. 5: Isn't the additional constraint a double penalty for radiation?

P7: Did you try to run the 14-parameter case with the same step size for the 7 parameters as you used in the 7-parameter case? You are using much smaller steps in the 14-parameter case for the parameters that you later identify as being the most dominant in the cost function (ENT, RHC, CT,. . .). (Again, just interested in your thoughts.)

P9 L7: What are the "successful parameter sets"? And how do the ensemble members differ from each other?

P9 L31: "extreme limits". This is probably linked to the choice of always perturbing towards the middle? Have you tried/thought of an algorithm design, where you would

lose the constraint of perturbing towards the middle and start the estimation closer to the default parameter values? This way the number of iterations might get smaller, or alternatively you could decrease the steps size (and algorithm termination criteria) and try to find the "exact" minimum of the cost function.

P11 L23-27: Not sure if this is a fair conclusion, the 14-parameter cases were much worse in performance than the 7-parameter cases (i.e. maybe a better constrained cost function could improve the 14-parameter converge at the cost of requiring more iterations/evaluations). P14 L11-15: I don't understand why you are comparing against the control only? Why not do this comparison against each parameter sets own individual coupled runs?

P15 L11: Your cost function is area based, why would the extra-tropics and tropics offset one another?

P15 L11: Aren't CT and CW almost the same as ICE_SIZE?

P17 L30: I would argue against drawing any conclusions based on numerical/toy model experimentation. In my opinion, parameter estimation/optimisation in GCMs is definitely not a smooth problem!

P18 L1: "models that appear similar." Is this only in the target criterion sense? There have to be differences between the models in some fields, no?

TECHNICAL COMMENTS P2 L19: double "parameters in the cloud scheme"

P4 L20: define S and O here already

P13 L1: "initial random" -> "initial extreme random"

P14 L4: "All four cases", a bit confusing, took me a while to understand it was 2 cases from 7-parameter cases + 2 from the 14-parameter cases.

P15 L19: "For these 6..." ?

P17 L23-24: "Given the sensitivity..." too long sentence, please rephrase.

Please also note the supplement to this comment:
http://www.geosci-model-dev-discuss.net/gmd-2016-305/gmd-2016-305-RC1-supplement.pdf

---

## Referee Comment (RC2) · P. Rayner (Referee) · 19 May 2017

This paper studies technical aspects of the calibration of parameters in a climate model using a range of observations. It extends previous work by including more parameters and more classes of observations. Its main concern is whether the process is technically feasible, that is whether the minimisation algorithms employed to find the maximum likelihood estimate of the parameters can converge and whether the converged values are reasonable. The paper is certainly in scope since it studies an important problem in climate science and investigates technical aspects of that problem.

I believe the paper needs substantial work before it can be published but it is possible that I am misunderstanding something quite simple about it and hence my concerns

might be irrelevant. At a practical level my concern is the temporal frequency of the observations being fitted. I didn't see this quoted in the text, presumably it is noted somewhere. In two extreme cases this will pose different kinds of problem for the paper.

1) High-frequency observations (e.g. daily) are used. In this case the sensitivity of the simulation and hence the cost function to the parameter is nearly arbitrary. A given simulation is one representation of the deterministic chaos of the model. The same perturbation in the parameters with a perturbed initial condition (correctly not included in the parameter estimation) might produce quite different sensitivities. The perturbation in the parameter presumably shifts the mean state of the simulation somewhat but the projection of this mean onto the time series might be very hard to see. In this case the gradient suggested by the derivative of the cost function might be a poor predictor of what happens when one actually searches in this direction. This looks like it might be happening but not for any technical reason but rather that the cost function is dominated by variations unpredictable by small parameter variations. This is a fascinating problem: What parts of the manifold in a chaotic system are legitimate targets for assimilation.

2) The other extreme case is that only long-term and large-scale observations are used, perhaps one observation per class. This would circumvent problem (1) but yield a quite different problem where the parameter estimation is under-determined. In this case we are back in the realm of conventional data assimilation where the use of prior information acts as regularisation as well as providing proper scaling etc for the parameters. Note that the authors are implicitly using some prior information by limiting the search space, it would be better to include this information within the probabilistic description of the problem (e.g. Tarantola 2005).

So, I'm not sure which or even whether these problems apply and clearly the authors need to describe their observational dataset more clearly but either way I believe some more work is needed.

There is also, I believe, one serious misunderstanding of the parameter estimation problem which has caused the authors to skip a step they actually can't avoid. On page 5 the authors state that it's not their problem to compute observational uncertainties which must come from those who generate the observations. I don't think this is correct. The observational uncertainty in a conventional estimation problem like this actually combines the error in the observation (difference between measured value and true value) and the difference between what the model should simulate for a given value of its inputs and what it actually does simulate. Here the inputs are parameters so the error likely concerns structural errors uncorrectable by any parameter setting. This is a task for the modeller and, unfortunately, not an easy one. In many problems like atmospheric inversion these model errors dominate the observational component. The authors should dsicuss and, possible, quantify this.

I also believe the authors need to talk some more about uncertainties in their parameters. Information on this is available from error propagation via the Jacobian from the observational covariance. This might be a simple explanation for the apparent equifinality.

Given these rather general concerns about the paper I will await a response before more detailed comments on the text. One concern that may affect any recalculations the authors may choose to do is the comment on page 8 about making sure the covariance is invertible. I agree this must be done, covariance matrices should be positive definite but wonder how singular matrices can appear in a correctly specified problem. Some covariance structures can yield near zero eigen-values but that should not be the case here.

---

## Author Comment (AC2) · 30 Jun 2017

**Response to reviewers**

Simon Tett,  Kuniko Yamazaki, Michael Mineter, Coralia Cartis and Nathan Eizenberg

We thank both reviewers for their time and effort in reading the paper and their comments. Below text from reviewers is in "normal" font. Our response in italic.

**Referee #1**

GENERAL COMMENTS The manuscript discusses using Gauss-Newton line-search  algorithm in optimising/tuning the atmospheric and coupled versions of Hadley Centre model. The optimisation uses a multi-criteria target, that includes 5 additional constraints to the authors' previous work. Optimisation is done for HadAM3-model by perturbing 7-/14-parameters. Similarly, parameter optimisation for HadAM3P-model is done by perturbing 7-/13-parameters. Additionally, tests are done for only perturbing a subset of the parameters at once. The optimisation seems to work well for the 7-parameter cases, but less so when the number of simultaneous parameter perturbations is increased. The results show the the GN line-search algorithm is able to optimise the parameter values in the training set, but also produces atmospheric models that can be used in coupled climate runs.

The manuscript is well written, and the results are well presented and interesting. The authors have done a good job in presenting numerous aspects of both the optimisation process as well as how the optimised parameters impact the model in climate runs. I have quite a few comments, but they probably won't require too much work. I am suggesting a "minor revision" for the manuscript.

*We thank the reviewer for her/his accurate summary of our paper, their positive remarks and useful comments.*

SPECIFIC COMMENTS Eq.3: Could you elaborate on "eˆi is the ith coordinate vector", the Nocedal and Wright book is not readily available.

*The e^i coordinate vector is the vector with $n$ components such that the ith entry of e^i is 1 while all remaining components are zero. It is used in the definition of approximate model derivative J_{ij} in equation (3). Please note that we have corrected a typo in J_{ij} in eq (3) and in the subsequent paragraph, which may have caused the confusion. We have also added a clarifying remark to the revised paper.  See page 5*

P5 L15: "...perturb towards the middle...". Perturbing always towards the centre is most likely a good idea, but would the algorithm converge if the optimal parameter values were at the edges of the "allowed" space? Just thinking about hypothetical cases where the default parameter values would be ill set. (Just interested in your thoughts about this, not necessary to add any text.)

*In theory this should be another 1st order estimate of the Jacobian so should not fundamentally change our results. Empirically our 7-parameter (and 14 parameter) cases were all started from parameters at the edge of the allowed range and the 7-parameter cases all converged. This*

*suggests that the computing the Jacobian this way is a reasonable thing to do.  The referee asks what if the optimal values were at the edge of the parameter space – The CT parameter often ended up on one extremum so this suggests that our approach would work in this case too.*

*If derivatives are sufficiently accurate (i.e. noise is small enough or can be controlled), optimization theory guarantees convergence both when the starting guess lies on the edge of the allowed set of solutions, and when the solution itself may lie on such boundaries.*

*Further research may well focus on estimating Jacobians more accurately for example by computing centred differences (which would be a 2nd order estimate) at the cost of another model evaluation and making perturbations large enough to avoid noise contamination.*

P5 L22: "several steps", how is this defined? Is the number of steps dynamic or static? Linked to this, could you make sure that you include the total number of steps also when you talk about how many iterations it took for the algorithm to converge/stop (e.g. P10 L3).

*We've added a  forward reference from P5L23-24 to help the reader. We define the number, and value, of the "line-search" steps when we describe the algorithm. To further  clarify: the number of steps in the line-search are predefined and fixed, so that the corresponding model evaluations can be computed in parallel.*

*We've included for the 7-parameter algorithm a statement on the number of evaluations needed – something already included for the other cases.  See P10L20*

Eq. 5: Isn't the additional constraint a double penalty for radiation?

*Referee is correct – it penalises the sum of SW & LW components from incoming value.*

*We've added a statement on P8L27  making this clear.*

P7: Did you try to run the 14-parameter case with the same step size for the 7 parameters as you used in the 7-parameter case? You are using much smaller steps in the 14-parameter case for the parameters that you later identify as being the most dominant n the cost function (ENT, RHC, CT,:::). (Again, just interested in your thoughts.)

*A good point but we haven't done this yet. We think further research beyond this demonstration paper would focus on better estimating the Jacobian matrix – for example, by tuning the step size as the algorithm progressed. The aim is to have small enough step sizes that differences are reasonable approximations to the derivative but not so small that the difference is strongly noise contaminated. One could also imagine running larger ensembles for some parameter perturbations or dropping those parameters that appear to have little impact on the Jacobian.*

P9 L7: What are the "successful parameter sets"? And how do the ensemble members differ from each other?

*These are the sets that converged to cost function values comparable with the standard case; the simulations were started with the same initial state but with different small perturbations.  A remark has been added to the revised manuscript. See P 9, L12-13*

P9 L31: "extreme limits". This is probably linked to the choice of always perturbing towards the middle? Have you tried/thought of an algorithm design, where you would lose the constraint of perturbing towards the middle and start the estimation closer to the default parameter values?

This way the number of iterations might get smaller, or alternatively you could decrease the steps size (and algorithm termination criteria) and try to find the "exact" minimum of the cost function.

*This is a good point – we have added a case where we started with the standard parameter set, please see revised discussion (P17). We also wanted to be able to find as many as possible parameter configurations that give good match to observations, and so we needed to initialize the algorithm from other starting guesses as well.*

P11 L23-27: Not sure if this is a fair conclusion, the 14-parameter cases were much worse in performance than the 7-parameter cases (i.e. maybe a better constrained cost function could improve the 14-parameter converge at the cost of requiring more iterations/evaluations).

*Table 5 shows that the for the HadAM3-14 case that the expected number of iterations is 8 which is indeed less than twice the expected number of iterations for the 7 parameter case (5.6).*

P14 L11-15: I don't understand why you are comparing against the control only? Why not do this comparison against each parameter sets own individual coupled runs?

*We are doing as the referee suggested. We have re-written the text to hopefully, make it clearer. (See P15L11)*

P15 L11: Your cost function is area based, why would the extra-tropics and tropics offset one another?

*We meant the OLR & RSR offset one another in the computation of net flux. Text has been rewritten to clarify. See P15L2-3*

P15 L11: Aren't CT and CW almost the same as ICE_SIZE?

*In broad terms the reviewer is correct that CT & CW have similar magnitude effects on the observations though there are differences. We think current text covers this.*

P17 L30: I would argue against drawing any conclusions based on numerical/toy model experimentation. In my opinion, parameter estimation/optimisation in GCMs is definitely not a smooth problem!

*We disagree with the referee. Neither HadAM3 nor HadAM3P are toy models – they both contain the same kind of structure as state-of-the-art GCMs albeit, for HadAM3 at low resolution compared to current models. We would be interested in evidence that model behaviour with respect to parameter choices is not smooth.*

P18 L1: "models that appear similar." Is this only in the target criterion sense? There have to be differences between the models in some fields, no?

*We have clarified that what we meant was the cost function. See P18L13*

TECHNICAL COMMENTS

P2 L19: double "parameters in the cloud scheme"

*Thank you, we deleted the second phrase.*

P4 L20: define S and O here already

*Thank you, we have done so.*

Vn 1 30.6.17

P13 L1: "initial random" -> "initial extreme random"

Thanks and made suggested change.

P14 L4: "All four cases", a bit confusing, took me a while to understand it was 2 cases from 7-parameter cases + 2 from the 14-parameter cases.

*Thanks for this point and have revised text. See P14L31*

P15 L19: "For these 6..." ?

Thanks, we have modified. We are now more explicit and hopefully text is now clearer. See P16L14

P17 L23-24: "Given the sensitivity..." too long sentence, please rephrase.

*We have revised the text and split sentence into two. See P18L24-26.*

Referee 2 (P. Rayner)
P. Rayner (Referee)

prayner@unimelb.edu.au

This paper studies technical aspects of the calibration of parameters in a climate mode using a range of observations. It extends previous work by including more parameters and more classes of observations. Its main concern is whether the process is technically feasible, that is whether the minimisation algorithms employed to find the maximum likelihood estimate of the parameters can converge and whether the converged values are reasonable. The paper is certainly in scope since it studies an important problem in climate science and investigates technical aspects of that problem

I believe the paper needs substantial work before it can be published but it is possible that I am misunderstanding something quite simple about it and hence my concerns might be irrelevant. At a practical level my concern is the temporal frequency of the observations being fitted. I didn't see this quoted in the text, presumably it is noted somewhere. In two extreme cases this will pose different kinds of problem for the paper.

*We have done our best to further explain and attempt to clarify here and in the paper the general issues raised by this review.*

1) High-frequency observations (e.g. daily) are used. In this case the sensitivity of the simulation and hence the cost function to the parameter is nearly arbitrary. A given simulation is one representation of the deterministic chaos of the model. The same perturbation in the parameters with a perturbed initial condition (correctly not included in the parameter estimation) might produce quite different sensitivities. The perturbation in the parameter presumably shifts the mean state of the simulation somewhat but the projection of this mean onto the time series might be very hard to see. In this case the gradient suggested by the derivative of the cost function might be a poor predictor of what happens when one actually

searches in this direction. This looks like it might be happening but not for any technical reason but rather that the cost function is dominated by variations unpredictable by small parameter variations. This is a fascinating problem: What parts of the manifold in a chaotic system are legitimate targets for assimilation.

2) The other extreme case is that only long-term and large-scale observations are used, perhaps one observation per class. This would circumvent problem (1) but yield a quite different problem where the parameter estimation is under-determined. In this case we are back in the realm of conventional data assimilation where the use of prior information acts as regularisation as well as providing proper scaling etc for the parameters. Note that the authors are implicitly using some prior information by limiting the search space, it would be better to include this information within the probabilistic description of the problem (e.g. Tarantola 2005).

So, I'm not sure which or even whether these problems apply and clearly the authors need to describe their observational dataset more clearly but either way I believe some more work is needed.

*We use approach 2 in the paper and have made various changes to the text to make this more explicit (sentence in the abstract (P1L4), introduction (P3L19) where we describe the observations (P7L28) and in the conclusions (P18L13). Though we did state in the modelling section the period we used this was far from clear and we thank Rayner for noting this.*

*In addition to bounds on the parameters, our regularization is explicitly imposed in the optimization scheme, and it is of Tichkonov-type, so as to make the resulting inverse problem well posed. The referee is correct that our results are conditional on the observations we choose. We chose a wide set of relevant observations available on large scales, which we justified in the text. Furthermore, our approach is justified by its conclusions that it leads to coupled models that do not require flux correction (or have small error in their SSTs).*

*We agree with the referee that optimising on the full spatial-temporal fields would be problematic. However, in terms of climate we think the right way to proceed would be to work with moments of the statistical distributions. Though estimating the covariance error structure would be challenging and would require extensive work from the data producers.*

There is also, I believe, one serious misunderstanding of the parameter estimation problem which has caused the authors to skip a step they actually can't avoid. On page 5 the authors state that it's not their problem to compute observational uncertainties which must come from those who generate the observations. I don't think this is correct. The observational uncertainty in a conventional estimation problem like this actually combines the error in the observation (difference between measured value and true value) and the difference between what the model should simulate for a given value of its inputs and what it actually does simulate. Here the inputs are parameters so the error likely concerns structural errors uncorrectable by any parameter setting.
This is a task for the modeller and, unfortunately, not an easy one. In many problems like atmospheric inversion these model errors dominate the observational component. The authors should discuss and, possible, quantify this.

*We disagree with the reviewer on this point. A priori it is not at all obvious that our attempts to produce a model with a smaller cost function than the standard model would fail. So attempting to produce a model with minimum error is the appropriate approach. We are not, in our case, in the business of data assimilation to carry out state estimation where bias correction can work (and can be tested enough times in forecast mode to have confidence in it). We are in the business of*

*producing climate models where minimising error is important. Thus observational error is the error in the actual observational estimates – something that currently is rarely done For example what is the current surface temperature of the Earth? Rayner is correct that model error could also be considered and we did consider this in our previous paper (Tett et al, 2013b) though feel there is a lot of subjectivity in that approach.*

*The challenge, as Rayner notes in his point 2, is deciding what is an appropriate cost function to be minimised. To do this would require much more research and the aim of our work was to demonstrate that it was feasible to calibrate O(10) parameters with a relatively simple optimisation/inverse approach.*

I also believe the authors need to talk some more about uncertainties in their parameters. Information on this is available from error propagation via the Jacobian from the observational covariance. This might be a simple explanation for the apparent equifinality.

*We thank the referee for this comment and have added a new subsection (2.8) to the methods section to show how we compute the parameter error covariance matrix. We then use this in subsection 3.2 to show that all ten 7-parameter optimised parameter sets are very different from one another.*

Given these rather general concerns about the paper I will await a response before more detailed comments on the text. One concern that may affect any recalculations the authors may choose to do is the comment on page 8 about making sure the covariance is invertible. I agree this must be done, covariance matrices should be positive definite but wonder how singular matrices can appear in a correctly specified problem. Some covariance structures can yield near zero eigen-values but that should not be the case here.

*Singular matrices arises when there is a linear relationship between rows or columns of the data matrix used to generate the covariance matrix. This can arise in two ways 1) a poor selection of observations. For example in our early trials we used pressure differences from the global-mean for all three of our regions. As one of the three regions can be linearly computed from the other two this lead to a non-invertible matrix. 2) The internal variability covariance matrix may not be invertible (or be near-singular) as it is estimated from model realisations. We verified that our covariance matrices are invertible.*

*Differences between revised and submitted paper are included as a supplement.*